# Regulation of antitumour CD8 T-cell immunity and checkpoint blockade immunotherapy by Neuropilin-1

Marine Leclerc[1], Elodie Voilin[1,7], Gwendoline Gros[1,7], Stéphanie Corgnac [1], Vincent de Montpréville[1,2], Pierre Validire[3], Georges Bismuth[4,5,6] & Fathia Mami-Chouaib[1]

Neuropilin-1 (Nrp-1) is a marker for murine $CD4^+FoxP3^+$ regulatory T (Treg) cells, a subset of human $CD4^+$ Treg cells, and a population of $CD8^+$ T cells infiltrating certain solid tumours. However, whether Nrp-1 regulates tumour-specific CD8 T-cell responses is still unclear. Here we show that Nrp-1 defines a subset of $CD8^+$ T cells displaying PD-1$^{hi}$ status and infiltrating human lung cancer. Interaction of Nrp-1 with its ligand semaphorin-3A inhibits migration and tumour-specific lytic function of cytotoxic T lymphocytes. In vivo, Nrp-1$^+$PD-1$^{hi}$ $CD8^+$ tumour-infiltrating lymphocytes (TIL) in B16F10 melanoma are enriched for tumour-reactive T cells exhibiting an exhausted state, expressing Tim-3, LAG-3 and CTLA-4 inhibitory receptors. Anti-Nrp-1 neutralising antibodies enhance the migration and cytotoxicity of Nrp-1$^+$PD-1$^{hi}$ $CD8^+$ TIL ex vivo, while in vivo immunotherapeutic blockade of Nrp-1 synergises with anti-PD-1 to enhance $CD8^+$ T-cell proliferation, cytotoxicity and tumour control. Thus, Nrp-1 could be a target for developing combined immunotherapies.

[1] INSERM UMR 1186, Gustave Roussy, EPHE, PSL, Faculté de Médecine-Université Paris-Sud, Université Paris-Saclay, 94805 Villejuif, France. [2] Centre chirurgical Marie-Lannelongue, Service d'Anatomie Pathologie, 92350 Le-Plessis-Robinson, France. [3] Institut Mutualiste Montsouris, Service d'Anatomie pathologique, 75014 Paris, France. [4] INSERM U1016, Institut Cochin, 75014 Paris, France. [5] CNRS, Unité mixte de Recherche 8104, 75014 Paris, France. [6] Université Paris Descartes, Sorbonne Paris Cité, 75006 Paris, France. [7] These author contributed equally: Elodie Voilin and Gwendoline Gros. Correspondence and requests for materials should be addressed to F.M.-C. (email: fathia.mami-chouaib@gustaveroussy.fr)

Cytotoxic T lymphocytes (CTL), predominantly expressing T-cell co-receptor CD8, play a major role in the anti-tumour immune response. To destroy malignant cells, CTL must first migrate to the tumour site, infiltrate tumour cell clusters, and then interact with malignant cells to achieve their cytotoxic functions after T-cell receptor (TCR) recognition of specific tumour peptide-major histocompatibility complex class I (MHC-I) complexes on target cells[1]. To perform this killing function, activated CTL are armed with various effector molecules, including pro-inflammatory cytokines, in particular IFNγ and TNF, and cytotoxic granules containing perforin and granzymes. However, cancer cells frequently escape CD8 T-cell recognition and reactivity. A decrease in cell surface expression of peptide/MHC-I complexes on cancer cells is one mechanism involved in this lack of functional activity. However, it now appears clear that engagement of inhibitory receptors such as CTLA-4, PD-1 and Tim-3, expressed on CD8+ tumour-infiltrating lymphocytes (TIL) with their respective ligands on target cells, is another critical constraint explaining the poor reactivity of these cells in the tumour immune context[2]. Moreover, PD-1 expression on CD8+ TIL seems to be a characteristic of clonally expanded CD8+ tumour-reactive T cells identified in cancer patients[3].

In this context, elucidating the mechanisms of CTLA-4 and PD-1 T-cell inhibitory signalling has led to development of promising cancer immunotherapy tools, including blocking monoclonal antibodies (mAb) targeting these so-called 'immune checkpoints'[4,5]. Yet, to be efficient, immune checkpoint blockade therapies require strong tumour infiltration by CTL whose activities are subjected to such inhibition. Indeed, the therapeutic benefit of the PD-1 blockade requires that tumours be infiltrated by CD8+ TIL strongly expressing the PD-1 receptor[6]. Under these conditions, CD8+ T-cell responses to tumour-specific antigens parallel tumour regression, and appear to be directly associated with clinical benefits of anti-PD-1 immunotherapy[3,7]. Unfortunately, only a fraction of cancer patients respond to PD-1 blockade and, among long-term responders, relapses are often observed after initial tumour regression. In this regard, a high proportion of tumours, referred to as immune deserts or cold tumours, are not infiltrated by immune cells and are thus poorly responsive to immunotherapy. Moreover, additional as yet undescribed inhibitory mechanisms inherent to CD8+ TIL and leading to tumour resistance likely exist.

In this light, we aimed to analyse the role of neuropilin-1 (Nrp-1) during antitumour CD8+ T-cell responses. Nrp-1 acts as a co-receptor for extracellular ligands, including secreted members of the class 3 semaphorin (Sema-3) family, isoforms of vascular endothelial growth factor (VEGF) and transforming growth factor β[8]. This transmembrane glycoprotein is involved in axon guidance and neuronal development[9,10]. More recently, a growing body of evidence suggests that Nrp-1 has multiple roles in the immune system and in T-cell responses. In mice, Nrp-1 is expressed at high levels on natural regulatory T cells, and its expression by Treg cells is responsible for reduced antitumour immunity[11,12]. Nrp-1 deficiency on murine FoxP3+ CD4+ Treg cells has been reported to delay tumour growth correlated with enhanced activation of tumour-infiltrating CD8+ T cells[13]. In humans, Nrp-1 is also expressed on a subset of CD4+ Treg cells in lymph nodes[14] and on CD4+ TIL, including suppressive Treg[15]. However, much less is known about the expression of Nrp-1 on CD8+ TIL and its involvement in regulating their functions. Nrp-1 has been found to be expressed by murine-tolerant self-reactive CD8+ T cells, and by both mouse and human melanoma-infiltrating CD8+ T lymphocytes[16]. Nevertheless, expression of Nrp-1 on CD8+ TIL has not been thus far systematically addressed, and its potential role in regulating

CTL activities and inhibiting antitumour T-cell functions is unknown.

Here we show that Nrp-1 is expressed on a subset of human CD8+ TIL in non-small-cell lung cancer (NSCLC) tumours and on murine CD8+ TIL from several tumour types. These T lymphocytes express high levels of PD-1, defining a subset of Nrp-1+PD-1hi TIL enriched with tumour-specific T cells. Anti-Nrp-1 blocking mAb restores CD8+ T-cell effector functions and optimise suppression of tumour progression induced by anti-PD-1. We conclude that Nrp-1 is an immune checkpoint that negatively regulates antitumour CD8+ T-cell immunity, and is thus a promising target for combination cancer immunotherapies.

## Results

**Nrp-1 is expressed on a subset of CD8+ TIL in human NSCLC.** Nrp-1 is expressed on CD4+ Treg cells in human lymph nodes and in TIL from colorectal cancer metastases[14,15]. However, little is known about its expression on CD8+ TIL. We first evaluated the expression of NRP1 transcripts in primary human lung tumours and autologous normal lungs. Quantitative real-time PCR (qRT-PCR) showed high expression levels of NRP1 mRNA in some lung tumour samples compared with the cognate normal lung (Supplementary Fig. 1a). The NSCLC tumour sample 8 was found to display a high increase in Nrp-1 mRNA expression, and NSCLC samples 1, 2 and 4 were found to display about a two fold expression increase compared with autologous healthy lungs. Human NSCLC were also found to express NRP2, such as tumour samples 1, 3 and 8 (Supplementary Fig. 1a), and certain Plexin (Plxn) A or D family member transcripts (Supplementary Fig. 1b), as well as genes encoding Nrp ligands of the Sema-3 family, such as Sema-3B, -3C, -3E and -3F (Supplementary Fig. 1c). At the protein level, we also found an expression of Nrp-1 on some epithelial tumour cells from freshly dissociated human NSCLC and on all the tested lung tumour cell lines (Supplementary Fig. 2a, b). These results were consistent with previous studies having shown that malignant cells frequently expressed Nrp-1, and in which Nrp-1 expression was associated with tumour progression[17–23].

We next focused on the expression of Nrp-1 protein in freshly isolated TIL. NSCLC tumours from 28 patients were analysed. Immunofluorescence analyses showed that Nrp-1 was expressed on a subset of human lung CD3+ TIL. In contrast, parallel analysis of peripheral blood lymphocytes (PBL) from lung cancer patients and healthy donors (HD) showed that circulating T cells did not express the receptor (Fig. 1a). Expression of Nrp-1 was observed on both CD8+ and CD4+ TIL, but with a higher frequency on CD8+ TIL (14.2 ± 2.1% vs 8.4 ± 0.9%). Moreover, expression of Nrp-1 correlated with the activation state of T lymphocytes, as it was more frequent on CD25+ and PD-1+ T cells from both CD8+ and CD4+ TIL subsets than on CD25− and PD-1− subsets (Fig. 1b, c). Indeed, 70.1 ± 8.3% of Nrp-1+ CD8+ TIL and 76.9 ± 4.2% of Nrp-1+ CD4+ TIL also expressed high levels of PD-1 (PD-1hi). Consistently, activation of HD PBL with immobilised anti-CD3 mAb induced expression of the protein (Supplementary Fig. 2c, d). In contrast, no obvious correlation between Nrp-1 and FoxP3 expression (p = 0.2578) was observed in CD4+ TIL, because equal percentages of Nrp-1+ T cells were found in both FoxP3+ and FoxP3− subsets, which showed similar percentages of PD-1hi cells (Fig. 1d, Supplementary Table 1). These results show that Nrp-1 is expressed on a subset of activated human CD4+ and CD8+ TIL displaying PD-1hi status in NSCLC tumours.

**Binding of Sema-3A to human Nrp-1 impairs T-cell functions.** Sema-3A, a secreted member of the Sema-3 family, is a

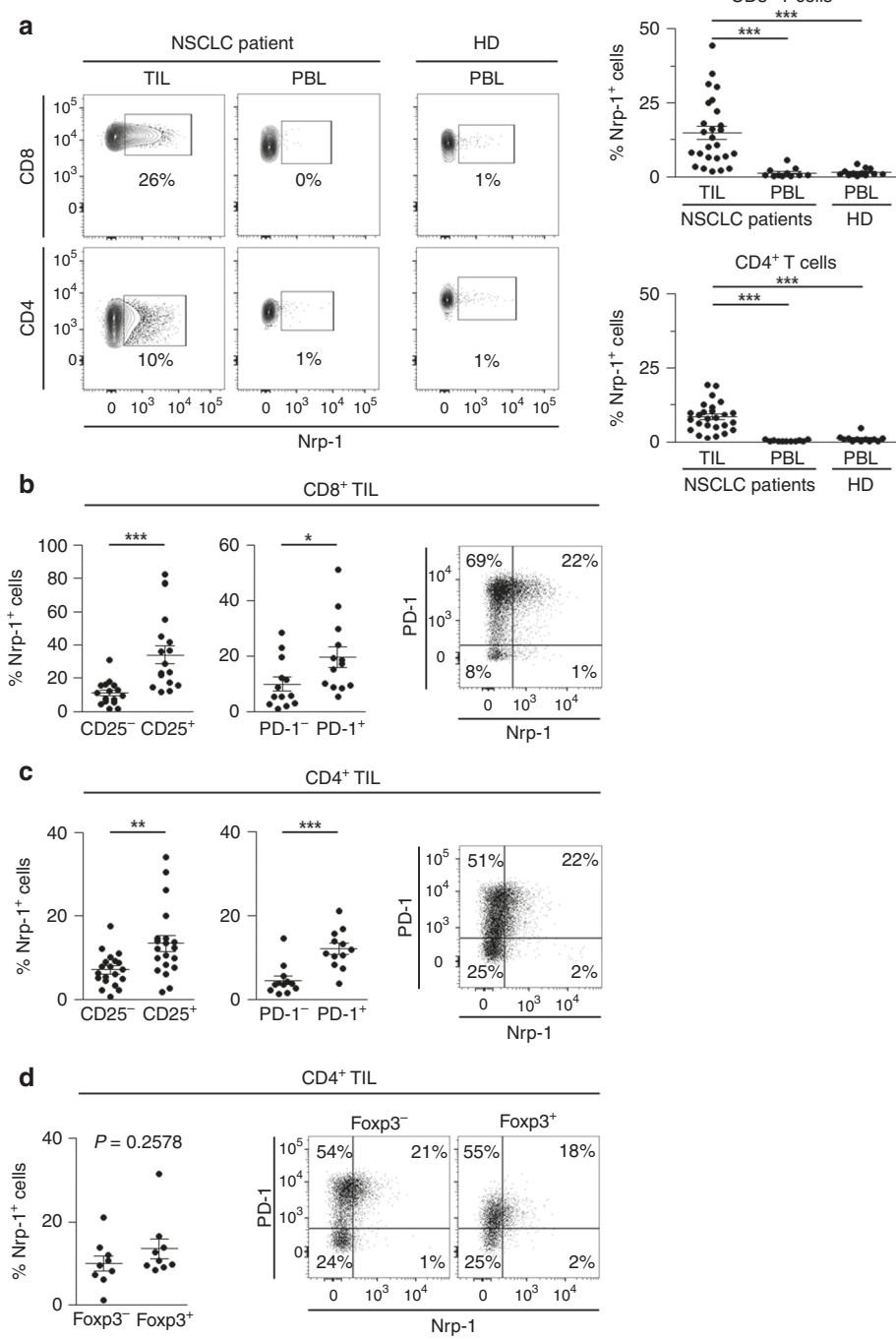

**Fig. 1** Expression of Nrp-1 and PD-1 on T cells infiltrating human NSCLC tumours. **a** Surface expression of Nrp-1 on CD4+ and CD8+ T cells from TIL and PBL of NSCLC patients, and compared with PBL from HD. Percentages of positive cells are indicated. Right: percentages of Nrp-1 on CD8+ and CD4+ T cells in TIL ($n = 28$) and PBL from NSCLC patients ($n = 11$) and HD ($n = 12$). **b** Expression of Nrp-1 in CD25 and PD-1 CD8+ T cells subsets from NSCLC TIL. Percentages of Nrp-1 among CD8+ TIL expressing or not CD25 ($n = 16$) or PD-1 ($n = 13$). Right: dot plot showing co-expression of Nrp-1 and PD-1 on CD8+ TIL from one representative patient. **c** Expression of Nrp-1 in CD25 and PD-1 CD4+ T cells subsets from NSCLC TIL. Percentages of Nrp-1 among CD4+ TIL expressing or not CD25 ($n = 20$) or PD-1 ($n = 12$). Right: dot plot showing co-expression of Nrp-1 and PD-1 on CD4+ TIL from one representative patient. **d** Expression of Nrp-1 in conventional and regulatory CD4+ T cells subsets. Percentages of Nrp-1 among CD4+ TIL expressing or not FoxP3 ($n = 9$). Right: dot plot showing co-expression of Nrp-1 and PD-1 on CD4+ TIL expressing or not FoxP3 from one representative patient. Means ± SEM one-way ANOVA test with Bonferroni correction **a** or two-tailed Student's unpaired $t$ test **b**, **c**, **d**. *$P < 0.05$; **$P < 0.01$; ***$P < 0.001$

well-known ligand of Nrp-1[9,10]. As human CD8+ TIL expressed substantial amounts of Nrp-1, this raises the question as to whether its interaction with Sema-3A contributes to the T-cell dysfunction often observed in the tumour microenvironment. To evaluate this hypothesis, we first used the IGR-Pub lung adenocarcinoma cell line and autologous CTL clone P62 established from TIL of a NSCLC patient[24]. Initial studies indicated that the IGR-Pub cell line, as well as fresh NSCLC tumours and several other human lung cancer cell lines, and to a lesser extent, normal bronchial epithelial cell line 16HBE, produced Sema-3A,

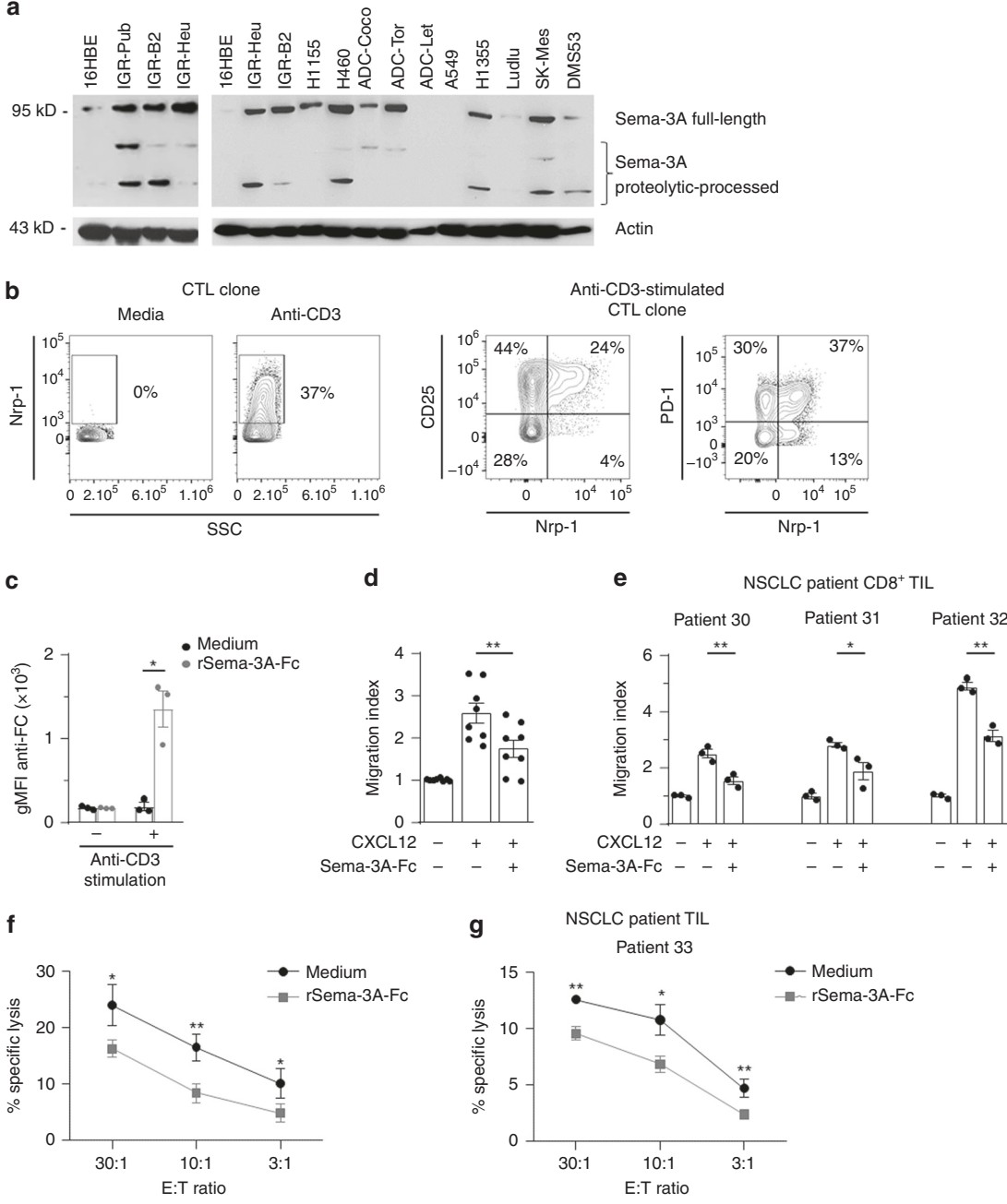

**Fig. 2** Expression of Sema-3A and Nrp-1 in human lung tumour cell lines and CTL clone. **a** Sema-3A expression in human lung tumour cell lines. Total protein extracts were analysed by western blot using anti-Sema-3A mAb. The 16HBE cell line was included as a control. Full length and proteolytically processed proteins are indicated. **b** Co-expression of Nrp-1 and CD25, and Nrp-1 and PD-1 on P62 T cells stimulated with immobilised anti-CD3. **c** The P62 clone was unstimulated or stimulated with anti-CD3, pre-incubated with Sema-3A-Fc, and then labelled with anti-human IgG-Fc secondary mAb. Results are gMFI mean ± SEM of triplicate samples. **d** Sema-3A-Fc inhibits CTL migration toward a CXCL12 gradient. The T-cell clone was stimulated with anti-CD3, pre-incubated with BSA or Sema-3A-Fc, and then seeded in the upper chambers of transwell plates and exposed to a gradient of CXCL12 loaded in the lower chambers. The number of T cells that had migrated into the lower chambers was determined. Results are mean chemotaxis index ± SEM of triplicate samples. **e** Sema-3A-Fc inhibits CD8+ TIL migration toward CXCL12. CD8+ TIL isolated from three human NSCLC tumours were stimulated with anti-CD3, pre-incubated with BSA or Sema-3A-Fc, and then seeded in the upper chambers of transwells and exposed to a CXCL12 gradient. Results are mean chemotaxis index ± SEM of triplicates. **f** Cytotoxicity of the CTL clone toward autologous tumour cells. The P62 clone was stimulated with anti-CD3, pre-incubated in medium or with Sema-3A-Fc. Cytotoxicity toward the cognate IGR-Pub cell line was determined. **g** Cytotoxic activity of freshly isolated TIL toward autologous tumour cells. TIL, freshly isolated from a NSCLC tumour, were stimulated with anti-CD3, pre-incubated in medium or with Sema-3A-Fc. Cytotoxicity toward freshly isolated autologous tumour cells was determined. Data shown for cytotoxic assay correspond to one of three independent experiments. Means ± SEM two-tailed Student's paired $t$ test **c**, one-way ANOVA test with Bonferroni correction **d**, **e** or two-way ANOVA test with Bonferroni correction **f**, **g**. *$P < 0.05$; **$P < 0.01$; ***$P < 0.001$. Source data are provided as a Source Data file **a**

as detected by western blot (Fig. 2a) and/or intracellular immu-nofluorescence staining (Supplementary Fig. 3a, b). Human fresh NSCLC tumours and tumour cell lines also expressed Sema-3F, another ligand of Nrp, as detected by intracellular staining (Supplementary Fig. 3c, d). In contrast, the expression of Nrp-1 on the P62 T-cell clone membrane varies according to its acti-vation state with feeder cells and IL-2. Consistently, stimulation with plastic-coated anti-CD3 mAb induced strong Nrp-1 expression on the surface of P62 T cells, most of which also expressed IL-2 receptor subunit CD25 and inhibitory receptor PD-1 (Fig. 2b). Therefore, we used the anti-CD3-stimulated P62 CTL clone to examine the consequences of the ligation to Nrp-1 of Sema-3A, for which recombinant (r) molecule is available, on the migratory behaviour and cytotoxic activity of these T lym-phocytes in vitro.

Previous immunofluorescence experiments indicated that P62 CTL bound rSema-3A-Fc (Fig. 2c) and expressed CXCL12 receptor CXCR4 (Supplementary Fig. 3e). Consequently, P62 T cells were able to migrate toward a gradient of rCXCL12 chemokine in in vitro transwell assays (Fig. 2d). We found that this T-cell migratory response was inhibited in the presence of rSema-3A-Fc (Fig. 2d). Similar results were obtained with TIL freshly isolated from three independent NSCLC patients showing that the interaction of human Nrp-1 with its soluble ligand Sema-3A inhibits T-cell migration toward CXCL12 gradient (Fig. 2e). This impaired T-cell migration was correlated with inhibition by rSema-3A-Fc of the cytotoxic activity of the P62 CTL clone toward the cognate IGR-Pub target cell line (Fig. 2f) and of freshly isolated NSCLC TIL toward autologous fresh tumour cells (Fig. 2g). These results indicate that Nrp-1 triggering by its soluble ligand of the Sema-3 family Sema-3A negatively regulates migration and cytotoxicity of Nrp-1$^+$ cytotoxic T cells, and suggest a T-cell inhibitory receptor function for Nrp-1.

**Nrp-1$^+$ CD8$^+$ TIL in B16F10 display an exhausted PD-1$^{hi}$ state**. To investigate the impact of Nrp-1 engagement with its ligand on CD8$^+$ TIL functions in vivo, we used C57BL/6 mice engrafted with B16F10 melanoma cells, which express the Nrp-1 soluble ligand Sema-3B, as shown by qRT-PCR analyses (Sup-plementary Fig. 4a) and confirmed by western blot (Fig. 3a). At day 15, tumours were removed and TIL were isolated and ana-lysed by flow cytometry. T cells from spleens and tumour-draining lymph nodes (TdLN) of the same mice were examined in parallel for Nrp-1 expression. Results showed that a large fraction of CD8$^+$ TIL expressed Nrp-1 (45.5% ± 2.5), as opposed to the very low percentages in spleens (1.9% ± 0.4) and TdLN (1.3% ± 0.1). The frequency of Nrp-1$^+$ T cells was also higher in CD4$^+$ TIL (37.7% ± 2.6), compared with spleens (15.7% ± 0.8) and TdLN (12% ± 0.4) (Fig. 3b). CD8$^+$ TIL isolated from engrafted lung tumour cell lines LL2 and TC-1, and the colon cancer cell line MC-38 also expressed high levels of Nrp-1, as opposed to spleens and TdLN (Supplementary Fig. 4b, c and 4d). Moreover, although most Nrp-1$^+$ CD4$^+$ cells from spleens and TdLN of B16F10-engrafted mice expressed the Treg cell marker FoxP3, Nrp-1$^+$ CD4$^+$ TIL included both FoxP3$^+$ and FoxP3$^-$ T cells (Fig. 3c). It should be noted that most, if not all, CD8$^+$ TIL displayed a CD44$^{high}$CD62L$^{low}$ phenotype, characteristic of effector/memory T (T$_{EM}$) cells, especially when they expressed Nrp-1 (Fig. 3d). Notably, as in humans, expression of Nrp-1 was dependent on the activation state of T cells. Indeed, stimulation of CD3$^+$CD8$^+$ mouse splenocytes with immobilised anti-CD3 induced Nrp-1 expression, which reached a plateau at 72 h (Supplementary Fig. 5a). It was inhibited by the NFAT inhibitor 11R-VIVIT, indicating that the TCR signalling pathway was likely involved (Supplementary Fig. 5b).

We then focused on induction of Nrp-1 on CD8$^+$ TIL during tumour progression. Expression of well-known inhibitory receptors PD-1, CTLA-4, Tim-3 and LAG-3 was monitored in parallel. Kinetic studies revealed that the percentage of Nrp-1$^+$ CD8$^+$ TIL increased during tumour growth, with similar induction of PD-1, LAG-3 and Tim-3 and, to a lesser extent, CTLA-4 (Supplementary Fig. 5c). The absolute number of T cells (i.e., the number of TIL per mg of tumour) expressing these inhibitory receptors also increased during melanoma progression, and reached a peak at day 14 after tumour implantation (Supplementary Fig. 5d). Remarkably, at day 14, all Nrp-1$^+$ CD8$^+$ TIL also expressed PD-1, and most of which also expressed LAG-3, Tim-3 and CTLA-4 (Fig. 4a). This Nrp-1$^+$ T-cell subset displayed a PD-1$^{hi}$ profile (Fig. 4a), with a strong correlation between expressions of the two cell surface molecules (Fig. 4b). Heat map analyses showed that this Nrp-1$^+$PD-1$^{hi}$ TIL subset included much higher percentages of T cells expressing other T-cell inhibitory receptors than the Nrp-1$^-$PD-1$^+$ and Nrp-1$^-$PD-1$^-$ subsets (Fig. 4c), with a much higher PD-1 geometric mean of fluorescence intensity (gMFI) than the Nrp-1$^-$PD-1$^+$ subset (Fig. 4d). They also indicated that the Nrp-1$^+$PD-1$^{hi}$ TIL subset included higher percentages of T cells expressing granzyme B and proliferation marker Ki-67, as well as higher gMFI of exhaustion-associated transcription factors such as NFATc1, IRF-4, Helios, Blimp-1 and T-bet, compared with the Nrp-1$^-$PD-1$^+$ subset (Fig. 4c). Nrp-1$^+$PD-1$^{hi}$ T cells also expressed high levels of *Batf*, *Cd244* and *Tigit* transcripts, the products of which were associated with dysfunctional T-cell status, but not *Egr2* mRNA (Supplementary Fig. 5e)[25]. It should be noted that most Nrp-1 was also found on FoxP3$^-$ CD4$^+$ T cells expressing PD-1, Tim-3 and CTLA-4, as well as Ki-67 (Supplementary Fig. 5f, g). These results indicate that Nrp-1 characterises an intra-tumoural CD8$^+$ T-cell subset displaying a highly activated PD-1$^{hi}$ status with co-expression of several T-cell inhibitory receptors, like CTLA-4, Tim-3 and LAG-3, involved in immune suppression during cancer diseases, and among which Nrp-1 may play an important role by repulsing activated T cells from the site of ongoing antitumour immune responses.

**Nrp-1 typifies activated-CD8$^+$ TIL with impaired activities**. Next, we investigated the specificity and functionality of these Nrp-1$^+$PD-1$^{hi}$ T cells. We first examined whether these lym-phocytes were enriched with T cells specific to melanoma-associated antigens (MAA). To do so, C57BL/6 mice engrafted with B16F10 melanoma cells were vaccinated with Trp2 plus gp100 antigenic peptides, together with the poly(I:C) adjuvant. Antigenic specificity of intra-tumoural CD8$^+$ TIL was then analysed with H-2-K$^b$-Trp2 and H-2-D$^b$-gp100 dextramers. Expression of Nrp-1 and PD-1 molecules on these TIL sub-populations was also monitored. Initial experiments showed that vaccinated mice more efficiently controlled tumour growth than unvaccinated mice (Supplementary Fig. 6a). Moreover, at day 14 after melanoma cell transplantation, tumours from vac-cinated mice showed increased absolute numbers of CD8$^+$ TIL expressing PD-1, Nrp-1 and Tim-3 and, to a lesser extent, CTLA-4 and LAG-3 (Supplementary Fig. 6b). More importantly, the percentage of Trp2-specific and gp100-specific CD8$^+$ TIL in the Nrp-1$^+$PD-1$^{hi}$ T-cell subset was over two-fold higher (8.8% ± 2.1 and 13.4% ± 1.5, respectively) than in Nrp-1$^-$PD-1$^+$ (2.9% ± 0.6 and 6.2% ± 0.7, respectively) TIL (Fig. 5a). In contrast, the Nrp-1$^-$PD-1$^-$ TIL subset showed very low percentages of MAA-specific T cells (0.9% ± 0.4 and 0.4% ± 0.1, respectively). Our results also revealed that ex vivo stimulation of Nrp-1$^+$PD-1$^{hi}$ TIL with autologous tumour cells induced higher percentages of IFNγ-producing T cells than in Nrp-1$^-$PD-1$^-$ and

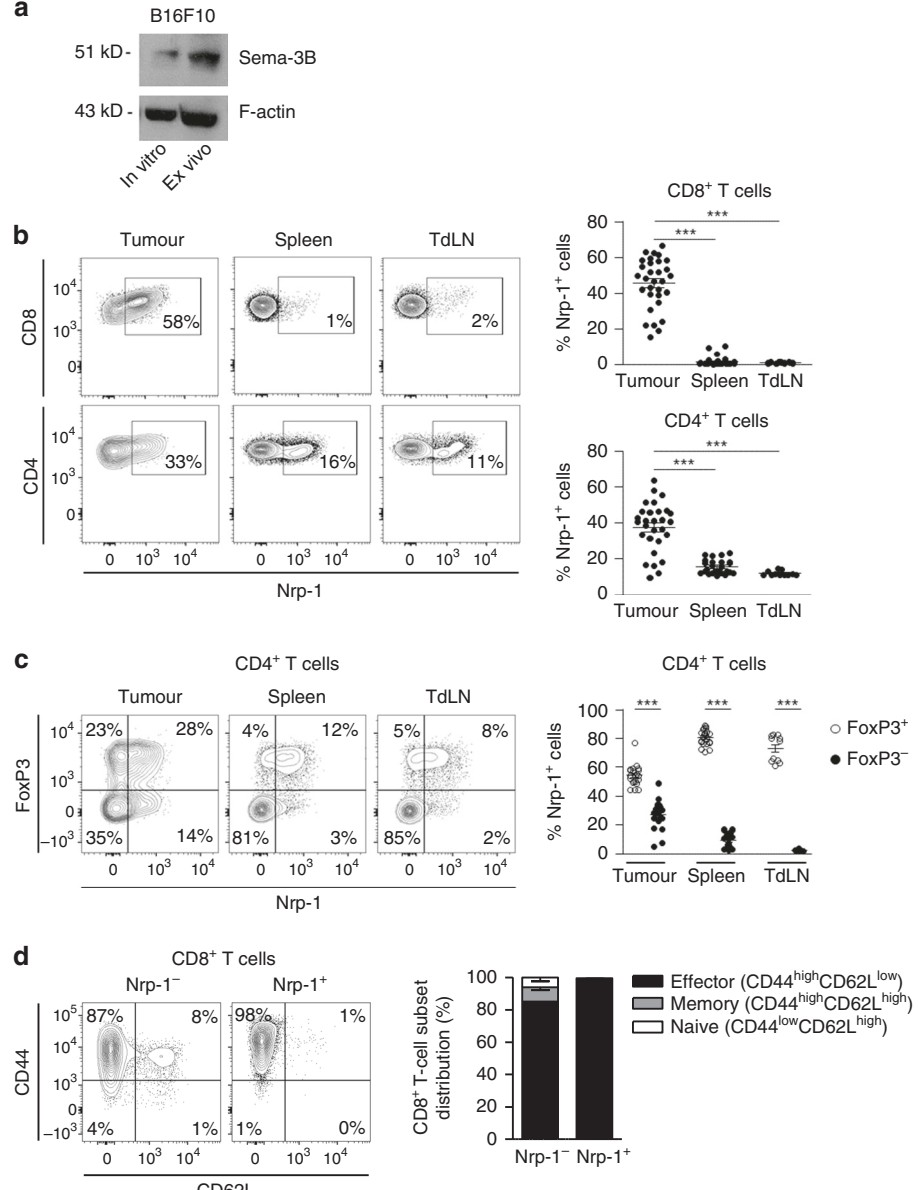

**Fig. 3** Expression of Sema-3B and Nrp-1 in B16F10 mouse melanoma model. **a** Expression of Sema-3B in B16F10 tumour cells. Total protein extracts from B16F10 cells cultured in vitro or isolated ex vivo from tumour grafts were analysed by western blot using anti-Sema-3B mAb. **b** Surface expression of Nrp-1 on CD4[+] and CD8[+] T cells infiltrating B16F10 melanoma engrafted in C57BL/6 mice. TIL from individual tumours were isolated at day 15 after tumour cell inoculation. T lymphocytes from spleens and TdLN of tumour-bearing mice were analysed in parallel. Percentages of positive cells are included. Right: percentages of Nrp-1[+] cells among CD8[+] and CD4[+] T cells in TIL ($n = 29$ and 31), splenocytes ($n = 29$ and 31) and TdLN ($n = 11$ and 16). **c** Expression of Nrp-1 and FoxP3 in CD4[+] T cells. T lymphocytes from tumours ($n = 20$), spleens ($n = 22$) and TdLN ($n = 11$) of B16F10 melanoma-bearing mice were analysed at day 15 by flow cytometry. Right: percentages of Nrp-1 among FoxP3[+] and FoxP3[-] CD4[+] T lymphocytes from B16F10. **d** Expression of CD44 and CD62L on Nrp-1[+] and Nrp-1[-] CD8[+] T cells from B16F10 TIL. Right: Distribution of naive, effector and memory T cells populations among Nrp-1[-] and Nrp-1[+] CD8[+] TIL ($n = 10$). Results are representative of 3–5 independent experiments. Means ± SEM one-way ANOVA test with Bonferroni correction **b** or two-way ANOVA test with Bonferroni correction **c**. *$P < 0.05$; **$P < 0.01$; ***$P < 0.001$. Source data are provided as a Source Data file **a**, **d**

the Nrp-1[-]PD-1[+] TIL subsets (Fig. 5b). Much stronger production of TNF was also found with Nrp-1[+]PD-1[hi] CD8[+] TIL compared with Nrp-1[-]PD-1[-] T cells, but in this case, percentages of TNF-producing T cells remained lower than in the Nrp-1[-]PD-1[+] TIL population. These results could be explained by a more advanced activation stage of the Nrp-1[+]PD-1[hi] CD8[+] TIL subset than the Nrp-1[-]PD-1[+] TIL subset. We then further assessed tumour reactivity of Nrp-1[+]PD-1[hi] and Nrp-1[-]PD-1[+] TIL by measuring CD107a expression, a marker usually used to evaluate the degranulation capacity of CTL. Our results showed

that ex vivo stimulation of TIL with the cognate B16F10 cell line induced much higher percentages of CD107a[+] cells among Nrp-1[+]PD-1[hi] and Nrp-1[-]PD-1[+] TIL populations than did the Nrp-1[-]PD-1[-] T-cell subset (Fig. 5c).

Next, we analysed the cytotoxic activity of CD8[+] TIL toward autologous melanoma cells in the absence or presence of neutralising anti-Nrp-1 and/or anti-PD-1 mAb, with an isotype-matched mAb as negative control. A 4 h chromium ([51]Cr) release assay revealed that CD8[+] TIL were poorly effective in killing B16F10 tumour cells, whether blocking mAb were

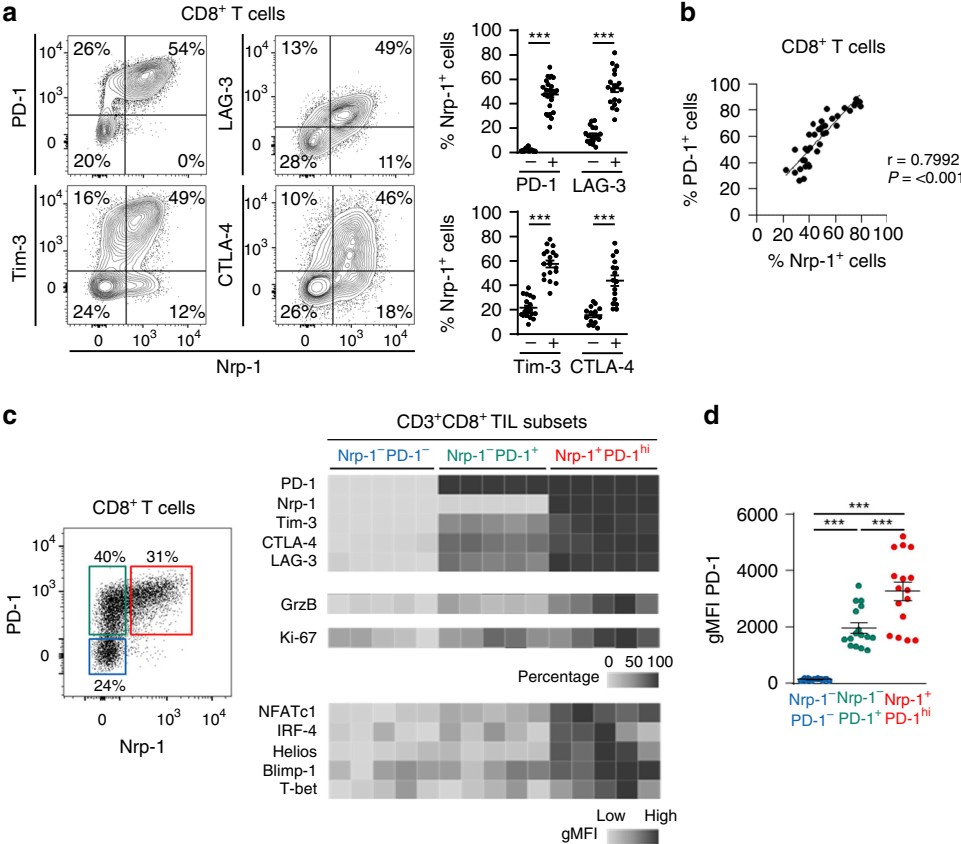

**Fig. 4** Expression of T-cell activation/exhaustion markers on the CD8+ TIL. **a** Expression of Nrp-1, PD-1, LAG-3, CTLA-4 and Tim-3 on CD8+ T cells from B16F10 TIL isolated at day 15. Right: percentages of Nrp-1 among CD8+ T cells expressing or not PD-1 ($n = 26$), Tim-3 ($n = 19$), LAG-3 ($n = 19$) or CTLA-4 ($n = 16$). **b** Expression of Nrp-1 on CD8+ T cells correlated with that of PD-1 ($n = 38$). **c** Expression of Nrp-1 on CD8+ TIL is restricted to a PD-1hi T-cell population. Left, Expression of Nrp-1 and PD-1 on CD8+ TIL; identification of three T-cell subsets: Nrp-1+PD-1hi, Nrp-1−PD-1+ and Nrp-1−PD-1−. Right: expression of exhaustion/activation markers, functional proteins and transcription factors in Nrp-1+PD-1hi, Nrp-1−PD-1+ and Nrp-1−PD-1− CD8+ T-cell subsets. Heat maps including percentages of cells positive for expression of PD-1, Nrp-1, Tim-3, CTLA-4, LAG-3, granzyme B (GrzB) and Ki-67, and gMFI for NFATc1, IRF-4, Helios, Blimp-1 and T-bet ($n = 5$). **d** gMFI of PD-1 on Nrp-1+PD-1hi, Nrp-1−PD-1+ and Nrp-1−PD-1− TIL subsets ($n = 16$). Results are representative of three to five independent experiments. Means ± SEM two-tailed Student's unpaired $t$ test **a** or one-way ANOVA test with Bonferroni correction **d**. *$P < 0.05$; **$P < 0.01$; ***$P < 0.001$. Source data are provided as a Source Data file **c**

present or not (Fig. 5d). In contrast, in a 12 h cytotoxicity assay, anti-Nrp-1 mAb, anti-PD-1 mAb, or a combination of both, strongly increased T-cell-mediated lysis. Notably, a combination of anti-Nrp-1 plus anti-PD-1 did not further increase target cell killing, suggesting that cytotoxicity is mainly mediated by the Nrp-1+PD-1hi TIL subset that includes most tumour-specific effector T cells (Fig. 5d). This cytotoxicity was correlated with upregulation of MHC-I molecules on the tumour cell surface (Fig. 5e). Indeed, parallel experiments in which we monitored H2-Kb and H2-Db expression, as well as PD-L1, on B16F10 cells co-cultured for 12 h with CD8+ TIL, showed increase of the three molecule expression, a phenomenon in which IFNγ secreted by T cells was likely involved. Consistently, we observed such an upregulation of H2-Kb/-Db surface expression after 12 h of stimulation of tumour cells with rIFNγ, and on melanoma cells collected from in vivo tumour grafts (Supplementary Fig. 6c). Moreover, neutralising anti-Nrp-1, anti-PD-1 and a combination of both mAb induced an increase in the percentage of CD8+ TIL expressing perforin after ex vivo stimulation with autologous tumour cells (Fig. 5f).

To further investigate the influence of Nrp-1 on CD8+ TIL functions, we also performed migration assays in the presence of anti-Nrp-1 and/or anti-PD-1-neutralising mAb of CD8+ TIL seeded for 2 h in upper chambers of transwell plates, with B16F10

target cells in the lower chambers. FACS analysis of T cells having migrated to the lower chambers in control conditions showed no difference between Nrp-1−PD-1−, Nrp-1−PD-1+ and Nrp-1+PD-1hi T-cell subsets (Fig. 5g). However, when TIL were pre-incubated with anti-Nrp-1 neutralising mAb, the migration index of Nrp-1+PD-1hi T cells was strongly increased. Whatever the TIL subset, no effect was observed with anti-PD-1 alone. Moreover, within the Nrp-1+PD-1hi T-cell subset, no further increase was observed with a combination of anti-Nrp-1 and PD-1 mAb. These results support the conclusion that Nrp-1 behaves as a true CD8 T-cell inhibitory receptor in vitro to impair the effector functions of antitumour CD8+ TIL.

**Nrp-1 blockade optimises tumour growth control by anti-PD-1.** The above experiments suggested that Nrp-1 is an immune checkpoint expressed by tumour-reactive T cells that could impair their functional activities following interaction with its ligand Sema-3B. To directly test this hypothesis in vivo and therefore investigate the potential of neutralising Nrp-1 via immunotherapeutic approaches, C57BL/6 mice were engrafted with B16F10 melanoma and treated at days 6, 8, 10, 12 and 14 with anti-Nrp-1, anti-PD-1, a combination of both, or an isotype control mAb. Both tumour volume and tumour weight were followed up. Results showed that intra-tumoural (i.t.)

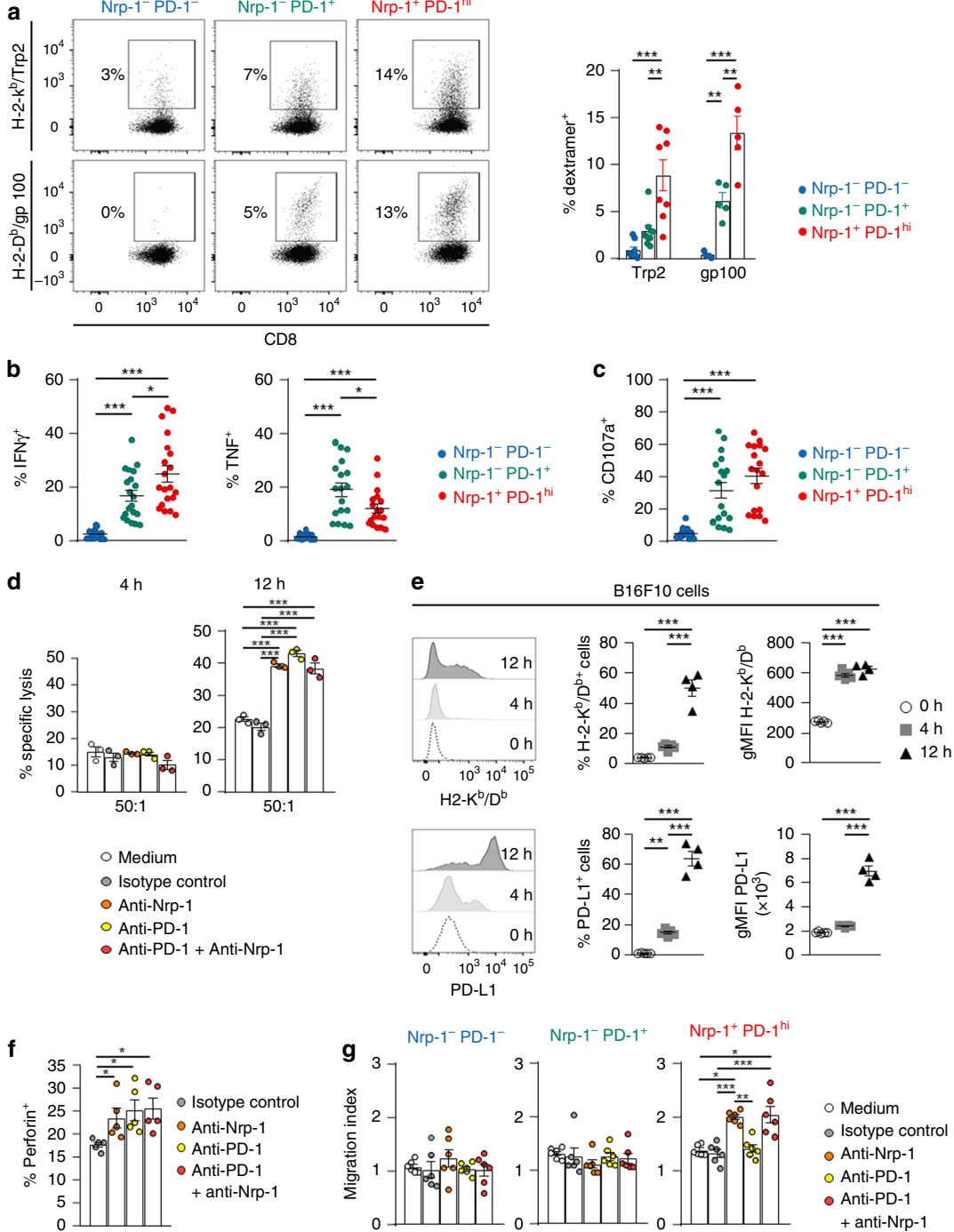

administration of anti-Nrp-1 mAb, or intra-peritoneal (i.p.) injection of anti-PD-1 mAb, inhibited tumour growth as compared with control treatment (Fig. 6a, b, Supplementary Table 2). In these experiments, the rational for using the intra-tumoural route to inject anti-Nrp-1 mAb was to reduce the quantity of administrated antibodies and potential side effects. Importantly, a combination of anti-Nrp-1 plus anti-PD-1 was clearly additive, with a much better control of tumour progression (Fig. 6a, Supplementary Fig. 6d). A strong reduction in tumour weight measured at the experiment endpoint was also observed (Fig. 6b). Similar experiments were conducted in C57BL/6 mice engrafted with MC-38 tumour cells and treated with anti-Nrp-1 mAb injected i.t. combined with anti-PD-1 injected i.p. A beneficial effect of anti-Nrp-1 treatment combined with anti-PD-1 could be observed (Fig. 6c, d). We checked in this model the effect of i.p. administration of anti-Nrp-1 on tumour growth and weight, and found that it was much more efficient than the i.t. route (Fig. 6c, d). Notably, when using the i.p. route of injection, anti-PD-1 plus anti-Nrp-1 combination had similar effects on tumour progression than anti-PD-1 plus anti-CTLA-4 (Supplementary Fig. 6e).

To further assess mechanisms involved in the therapeutic effect of anti-Nrp-1 plus anti-PD-1 combination, we analysed CD8+ TIL from B16F10 tumours of the different groups of mice used above. Results showed much larger CD8+ T-cell infiltrates in tumours obtained from mice treated with the anti-Nrp-1 plus anti-PD-1 combination than from mice treated with each mAb alone or the isotype control (Fig. 6e). This finding might be owing to the specific capacity of anti-Nrp-1 mAb to restore T-cell

**Fig. 5** The Nrp-1+PD-1hi TIL subset is enriched with activated antigen-specific CD8+ T cells. **a** Staining of CD8+ TIL with dextramers. C57BL/6 mice were engrafted with B16F10 and then vaccinated with Trp2 and gp100 peptides. On day 15, TIL were isolated from tumours. Right: Percentages of Trp2 ($n = 8$) and gp100 ($n = 5$) dextramer-positive T cells among Nrp-1+PD-1hi, Nrp-1−PD-1+ and Nrp-1−PD-1− CD8+ TIL. **b** Expression of IFNγ ($n=20$) and TNF ($n=18$) in Nrp-1+PD-1hi, Nrp-1−PD-1+ and Nrp-1−PD-1− CD8+ T cells. TIL were stimulated for 4 h with autologous tumour cells. **c** Degranulation of CD8+ TIL. TIL were stimulated with autologous tumour cells; then, T-cell subsets were analysed for expression of CD107a ($n = 18$). **d** Cytotoxicity of freshly isolated CD8+ TIL. CD8+ TIL were pre-incubated in medium or with anti-Nrp-1, anti-PD-1 or a combination of both mAb; then, cytotoxicity toward autologous tumour cells was determined. **e** Increase in MHC-I and PD-L1 expression on tumour cells co-cultured with autologous CD8+ TIL. Kinetic studies of H-2-Kb/-Db and PD-L1 expression on B16F10 co-cultured with CD8+ TIL. Expression profiles (left), percentages of positive cells (middle) and gMFI (right) of MHC-I (upper panels) and PD-L1 (lower panels) are shown. **f** Expression of perforin in CD8+ T cells. TIL were stimulated with autologous tumour cells in the absence or presence of neutralising anti-Nrp-1, anti-PD-1 or anti-Nrp-1 plus anti-PD-1 then, T-cell subsets were analysed for expression of perforin ($n = 5$). **g** Anti-Nrp-1 re-establishes migration of Nrp-1+PD-1hi T cells toward B16F10. TIL were pre-incubated in medium or with neutralising anti-Nrp-1, -PD-1, a combination of both mAb or an isotype control. Cells were seeded in the upper chambers of transwells and then exposed to a gradient of B16F10 supernatant, enriched in Sema-3B, loaded in the lower chambers. The numbers of Nrp-1+PD-1hi, Nrp-1−PD-1+ and Nrp-1−PD-1− T cells that had migrated were determined by flow cytometry. Results are representative of three independent experiments. Means ± SEM one-way ANOVA test with Bonferroni correction **a–g**. *$P < 0.05$; **$P < 0.01$; ***$P < 0.001$

migration to the tumour site. Increased CD8+/CD4+ Treg ratios were also observed in this group of mice compared with control mice (Fig. 6f). These CD8+ TIL expressed much higher levels of terminally differentiated effector T-cell marker KLRG1, as well as increased levels of proliferation marker Ki-67 and of serine protease granzyme B (Fig. 6g). To provide further mechanistic insights on these mAb combination effects, TIL from MC-38 tumours were isolated, the number of CD8+ T cells was determined and the cytotoxic activity toward autologous tumour cells was assessed. Results showed a much larger CD8+ T-cell infiltration in tumours obtained from mice treated with the anti-Nrp-1 i.p. plus anti-PD-1 combination than from mice treated with each mAb alone or the isotype control (Fig. 6h). Moreover, CD8+ TIL from mice treated anti-Nrp-1 i.p. plus anti-PD-1 combination or anti-Nrp-1 i.t. plus anti-PD-1 combination kill more efficiently MC-38 tumour cells ex vivo than TIL from mice treated with anti-Nrp-1 i.t. or i.p. alone, anti-PD-1 alone or isotype control mAb (Fig. 6i). These results correlated with an increase in the number of CD8+ TIL expressing granzyme B from mice treated i.p. with a combination of anti-Nrp-1 and anti-PD-1 than mice treated with anti-Nrp-1 alone, anti-PD-1 or isotype control mAb (Fig. 6j). Overall, these results demonstrate that anti-Nrp-1 plus anti-PD-1 in vivo treatment enhances tumour infiltration by CD8+ TIL, with increased proliferative and killing capacities, leading to strong tumour regression. They also further emphasise the therapeutic potential and benefits of the anti-Nrp-1/anti-PD-1 combination in cancer immunotherapy.

## Discussion

Nrp-1 is essential in axonal guidance, through its capacity to bind class 3 chemorepulsive semaphorin proteins[9,26], and in angiogenesis, via its interaction with VEGF and VEGF receptors (VEGFR)[27,28]. Nrp-1 is also involved in cardiovascular and neuronal development, cell migration, immunity and cancer[8,29,30]. Indeed, tumour cells frequently express semaphorins and their receptors neuropilins and plexins, which can regulate malignant cell behaviour and contribute to malignant potential[30]. Nrp-1 is also highly expressed on tumour vasculature, functioning as a mediator of tumour initiation and progression, associated with poor clinical outcome[8]. For instance, high expression of Nrp-1 observed in lung cancer correlates with invasive capacity and short disease-free survival[31]. Moreover, cancer cells often produce secreted members of the semaphorin family, including Sema-3A and Sema-3B, also contributing to tumour escape from the immune response[30,32]. We now report, in both human NSCLC tumours and murine B16F10 melanoma, that Nrp-1 delineates a particular subset of CD8+ TIL, enriched with tumour antigen-specific T lymphocytes, and also expressing high levels of the PD-1 inhibitory receptor. In particular,

we show that Nrp-1 works as a negative regulator of antitumoural activities of this CTL subset, and that blocking of this receptor in vivo strongly improves tumour regression elicited by anti-PD-1 immunotherapy.

Multiple roles for neuropilins and semaphorins in the immune system have emerged in recent years[33,34]. Sema-3A has been reported to inhibit primary human T-cell proliferation and cytokine production under anti-CD3 plus anti-CD28 stimulation[32]. It also promotes T-cell apoptosis[35] and inhibits non-specific cytotoxic activity of NK cells in mixed lymphocyte cultures[32]. The influence of Sema-3A on T-cell migration has also been studied. In particular, this chemorepulsive molecule has been reported to inhibit immune cell migration and response of human T cells to chemokine gradients[36,37]. Chemorepulsive effects of Sema-3A on human thymocytes have been reported, as well as inhibition of T-cell migratory responses triggered by chemokine CXCL12[38,39]. Accordingly, we found that soluble Sema-3A binds to Nrp-1 molecules expressed on the surface of human lung tumour-specific CTL and inhibits their migratory capacity to a gradient of CXCL12 chemokine. Importantly, our results also showed that TCR-mediated killing of autologous human lung cancer cells was inhibited in the presence of Sema-3A, highlighting a possible inhibitory role of the Nrp-1/Sema-3A axis in CTL functions.

Nrp-1 is usually not expressed by resting T cells. However, its expression is triggered after T-cell activation, suggesting that Nrp-1 is an additional T-cell activation biomarker[40]. Regulation of Nrp-1 expression during adaptive immune responses is likely an essential element for understanding its physiological role. In this context, activated T cells derived from inflammatory environments were described as expressing this surface molecule[16]. Indeed, Nrp-1 was highly induced on CD8+ T lymphocytes engaging self-antigen, including human melanoma TIL, supporting the hypothesis that it may correspond to a potential biomarker for dysfunctional T cells. We show here that Nrp-1 is co-expressed with CD25 and PD-1 on both CD4+ and CD8+ TIL from human NSCLC. Moreover, on CD8+ TIL, this expression is restricted to a PD-1hi T-cell subset. Various populations of CD8+ TIL have been recently described based on PD-1 expression levels: negative (PD-1N), intermediate (PD-1int) and high (PD-1T)[41]. Interestingly, we found a very good correlation between PD-1 and Nrp-1 expression (see Fig. 4b). Thus, the aforementioned PD-1T TIL subset likely corresponds to cells expressing the highest levels of PD-1 and Nrp-1 in our study. This is a crucial assumption, as this subset has been recently reported to correspond to an exhausted TIL subset[41]. Accordingly, in the present study, we found that Nrp-1+PD-1hi TIL express transcription factors Helios, Blimp-1, IRF-4 and NFATc1,

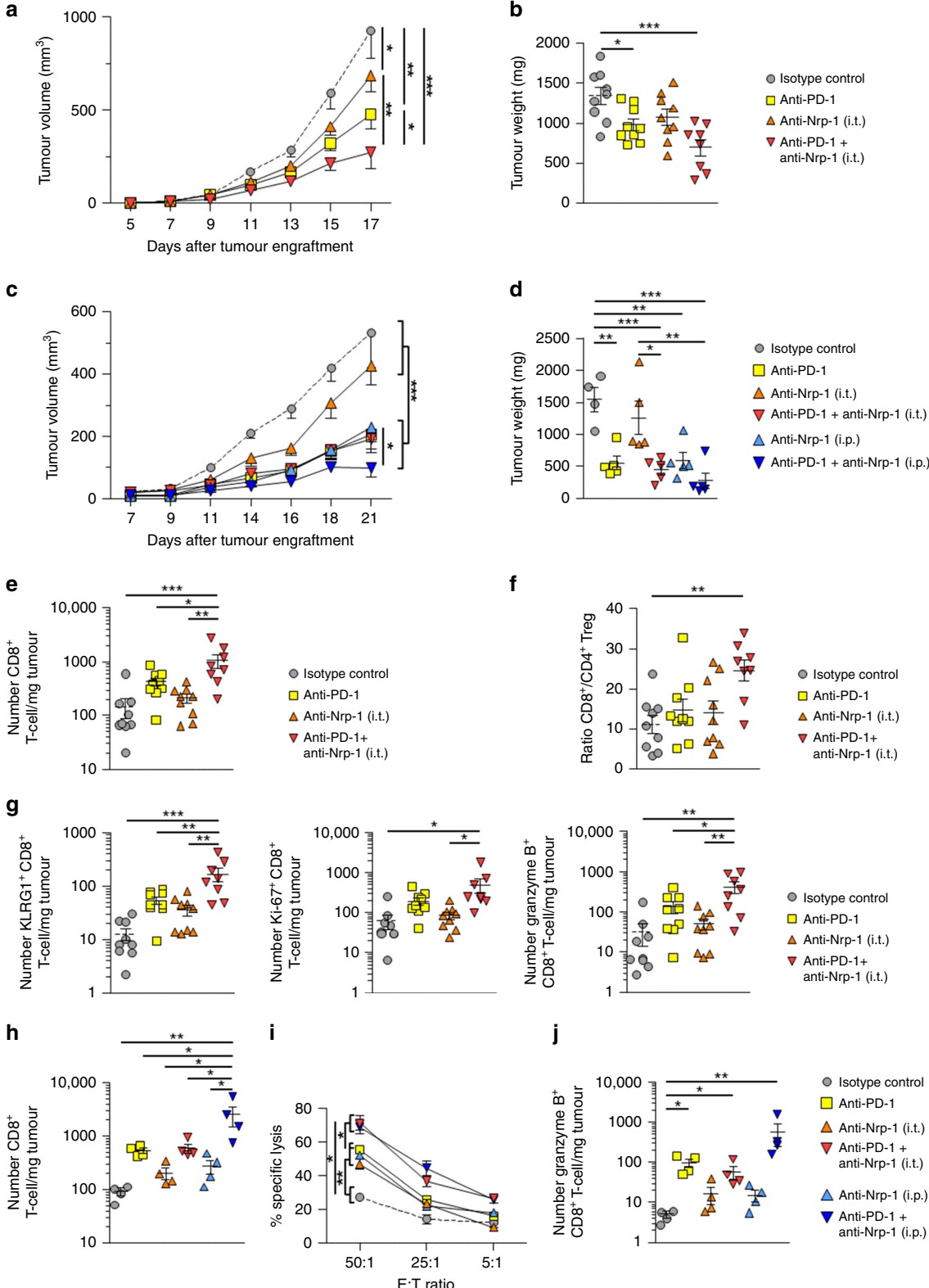

associated with activation/exhaustion[42–45]. Co-expression of Nrp-1 and Helios on CD4+ T cells was also observed on TIL from human liver metastases of colorectal cancer[15]. Another characteristic of PD-1[T] TIL is that they display high tumour recognition capacity and are predictive of responses of human NSCLC patients to PD-1 blockade[41]. Likewise, we found, in B16F10 tumours, that a significant percentage of Nrp-1+PD-1[hi] CD8+

TIL strongly expressed granzyme B and cell proliferation marker Ki-67, and secreted high levels of IFNγ. Moreover, this Nrp-1+PD-1[hi] T-cell subpopulation was the most strongly enriched with MAA-specific CD8+ T lymphocytes. We therefore assume that Nrp-1 can be used as a marker for identifying this important CD8+ tumour-reactive TIL population. Overall, these findings suggest that, in an antitumour immune response context,

**Fig. 6** Anti-Nrp-1 mAb combined with anti-PD-1 improve tumour progression control. **a** Mice were engrafted with B16F10 and then treated i.p. with anti-PD-1, i.t. with anti-Nrp-1, or a combination of both mAb injected in parallel at each site at days 6, 8, 10, 12 and 14 after tumour inoculation. Tumour volumes are given as means ( ± SEM) of nine mice/group. Mice were killed at day 17. Data are means of two independent experiments out of three. **b** Weight of tumours from mice untreated and treated with blocking mAb in **a**. at day 17. **c** C57BL/6 mice were engrafted with MC-38 tumour cells and then treated i.p. with anti-PD-1, i.t. or i.p. with anti-Nrp-1, or a combination of anti-PD-1 (i.p.) plus anti-Nrp-1 (i.t. or i.p.) injected at days 4, 7, 10 and 16 after tumour inoculation. Tumour volumes are given as means ( ± SEM) of 4–5 mice/group. **d** Weight of tumours from mice untreated and treated with blocking mAb in **c**. recovered at day 21. **e** Numbers of CD8$^+$ T cells per milligram of B16F10 tumour were determined at day 17. **f** Ratio of CD8$^+$/CD4$^+$ Treg cells in B16F10 tumours from mice treated with blocking mAb or an isotype control. **g** Numbers of KLRG1$^+$, Ki-67$^+$ and granzyme B$^+$ CD8$^+$ T-cell per milligram of B16F10 tumour from mice treated with blocking mAb or an isotype control. **h** Numbers of CD8$^+$ T cells per milligram of MC-38 tumour were determined at day 21. **i** Cytotoxic activity of freshly isolated CD8$^+$ TIL toward MC-38 tumour cells. CD8$^+$ TIL were isolated at day 21 in **c**. and then cytotoxicity toward autologous tumour cells was determined. **j** Absolute cell counts of granzyme B$^+$ CD8$^+$ T cells from MC-38 TIL. Numbers of T-cell subsets per milligram of tumour from mice treated i.p. with anti-PD-1, i.t. or i.p. with anti-Nrp-1, or a combination of anti-PD-1 (i.p.) plus anti-Nrp-1 (i.t. or i.p.) mAb. Means ± SEM two-way ANOVA test with Bonferroni correction **a**, **c**, **i** or one-way ANOVA test with Bonferroni correction **b**, **d**, **e**, **f**, **g**, **j**. *$P < 0.05$; **$P < 0.01$; *** $P < 0.001$

Nrp-1 is a very late T-cell activation marker and that its co-expression with high levels of PD-1 might be another characteristic of exhausted antigen-experienced CD8$^+$ TIL.

Expression of CTLA-4, PD-1 and Tim-3 molecules was associated with T-cell exhaustion in chronic viral infections and tumour progression[46–50]. These T-cell inhibitory receptors are important for regulating immune responses in peripheral tissues and maintaining self-tolerance. Along the same lines, Nrp-1 is induced on tolerant self-reactive CD8$^+$ T cells expressing known regulatory receptors[16]. In the present report, we show that Nrp-1 is often associated with a broad panel of inhibitory receptors, such as CTLA-4, Tim-3 and LAG-3, and characterizes a dysfunctional PD-1$^{hi}$ CD8$^+$ TIL subpopulation. This particular phenotype, characteristic of T cells infiltrating an inflamed tumour micro-environment, may further explain the paradox of tumour progression, despite the presence of an ongoing T-cell response toward malignant cells. We also show here that anti-Nrp-1 neutralising mAb, in particular when associated with anti-PD-1, re-established ex vivo the functionality of these T cells, such as increase of perforin expression and TCR-mediated cytotoxicity toward the cognate tumour. Remarkably, anti-Nrp-1 also restore the migratory capacity of Nrp-1$^+$PD-1$^{hi}$ T cells, likely by blocking the interaction of Nrp-1 with its ligands Sema-3A or Sema-3B. Anti-PD-1 blocking mAb have no such effect. Semaphorins have been reported to govern cell migration by regulating integrin-mediated adhesion and actin cytoskeleton[51]. A similar process may occur in tumours to explain how Nrp-1 signalling specifically affects T-cell migratory potential, possibly downstream plexin-A, which forms stable heterodimers with Nrp-1 and acts as a signal-transducing module for the Sema-3/Nrp-1/Plxn-A complex at the plasma membrane[52].

It is becoming increasingly clear that Nrp-1 and its ligands Sema-3A/−3B play important roles during the effector phase of various immune processes, including antitumour T-cell responses. A growing body of evidence indicates that these molecules are commonly expressed by cancer cells and immune cells, and that they may affect antitumour immune responses. In this regard, it has been reported that LKB1 protein kinase[53] and the ABL1 inhibitor Imatinib[54] inhibit tumour growth and angiogenesis by enhancing Nrp-1 degradation. Moreover, synthetic anti-Nrp-1 peptides[31] and blocking anti-Nrp-1 mAb have been developed[55] for targeting tumour angiogenesis. It has also been reported that inhibition of Sema-3C binding to Nrp-1 with small molecules attenuates tumour growth in prostate cancer[56]. In this context, our work is a building block, since we now show that Nrp-1 negatively influences CD8 T-cell immunity and responses to anti-PD-1 cancer immunotherapy. Indeed, our in vivo experiments have revealed that, like the anti-PD-1 blockade, anti-Nrp-1 immunotherapy is able to inhibit melanoma progression in C57BL/6 mice. Moreover, a combination of the two antibodies is more efficient at reducing tumour growth. In contrast, combination of anti-Nrp-1 plus anti-PD-1 did not have an additive effect on T-cell-mediated cytotoxicity ex vivo. The mechanism of the differential outcome in these two experimental systems is likely associated with the capacity of anti-Nrp-1 to enhance T-cell migration and recruitment within the tumour. Consistently, reduced tumour growth in vivo was associated with enhanced tumour infiltration by CD8$^+$ effector T cells, with an increase in the CD8$^+$/CD4$^+$ T-cell ratio. Moreover, CD8$^+$ TIL from mice treated i.p. with anti-Nrp-1 plus anti-PD-1 expressed higher levels of granzyme B and mediated stronger cytotoxic activity toward autologous MC-38 tumour cells than TIL from mice treated with each mAb alone. This is an important finding, as the limited success of anti-PD-1 cancer immunotherapy is often associated with weak tumour infiltration by specific CD8$^+$ T cells[57,58]. Concomitant blockade of Nrp-1 and PD-1 could remedy this cold-tumour situation, thus providing an immunotherapeutic strategy for further improving specific immune responses during cancer disease.

Together, these results demonstrate that blockade of Nrp-1 on CD8$^+$ T cells holds therapeutic potential and provides a strong rational for exploiting Sema and Nrp inhibitors as promising drugs for cancer treatment. Anti-Nrp-1 plus anti-PD-1 combination immunotherapy would allow optimisation of tumour infiltration by CD8$^+$ T cells and CTL reactivity toward target cells thereby improving immune protection and responses to immune checkpoint blockade. Furthermore, our results suggest that Nrp-1 expression on CD8$^+$ TIL could be used as a potential biomarker to predict response to anti-PD-1 treatment of cancer patients. In this regard, it has been recently reported that PD-1$^{high}$ CD8$^+$ T-cell pool was strongly predictive for response and survival in NSCLC treated with PD-1 blockade[41]. This Nrp-1$^+$PD-1$^{high}$ CD8$^+$ TIL subset is enriched with CD8$^+$ T cells recognising cancer cells and thus could be used for identifying patients who would respond to antibody combination therapies. Moreover, expression of Nrp-1 on PD-1$^{high}$ CD8$^+$ TIL may also permit easier quantification of this T-cell pool displaying tumour reactivity. Future studies should determine whether targeting additional inhibitory receptors on this TIL subset would further increase their antitumour activities. It would also be interesting to determine whether this Nrp-1$^+$PD-1$^{high}$ CD8$^+$ TIL subset is influenced by PD-1 blockade in human cancer samples and to validate Nrp-1 expression on TIL as a predictive biomarker for response to cancer immunotherapy.

## Methods

**Human PBL, lung tumours and freshly isolated lung TIL.** Healthy donor blood samples were collected from the French blood bank (Etablissement français du Sang (EFS); agreement number N°12/EFS/079), and patient blood samples were collected from Gustave Roussy. All patients were suffering from advanced and inoperable NSCLC stage IIIB/IV. Immune monitoring in the blood of patients was

approved by the Kremlin-Bicêtre Hospital Ethics Committee (no. 110–08; ID RCB: 2008-A01171–54), and Declaration of Helsinki protocols were followed.

Fresh NSCLC tumours were obtained from the Centre chirurgical Marie Lannelongue and the Institut mutualiste Montsouris. RNA was immediately extracted with TRIzol reagent (Invitrogen), reverse-transcribed and then subjected to qRT-PCR.

For freshly isolated TIL, human lung tumours were dissociated mechanically and enzymatically using a tumour dissociation kit (Miltenyi Biotec 130-095-929). Mononuclear cells were then isolated by a Ficoll-Hypaque gradient, and epithelial tumour cells were isolated using a tumour cell isolation kit (Miltenyi 130-108-339). Human experiments were approved by the Institutional Review Board of the Gustave Roussy Institute. All recruited healthy volunteers and patients provided written informed consent.

**Derivation of the P62 CTL clone and IGR-Pub tumour cell line**. NSCLC cell line IGR-Pub was derived from the tumour specimen of patient Pub adenocarcinoma in one of our laboratory[24]. Autologous CTL clone P62 was derived from TIL of the same patient[59]. T-cell clone P62 was stimulated every month with irradiated autologous IGR-Pub tumour cells and irradiated allogeneic Laz509 EBV-transformed B cells in Rosewell Park Memorial Institute-1640 medium supplemented with 10% human AB serum and rIL-2[24].

The allogeneic NSCLC cell lines IGR-B2, IGR-Heu, ADC-Coco, ADC-Tor and ADC-Let were derived from tumour specimens in one of our laboratories[59]. A549 (ADC), SK-Mes, Ludlu (SCC), DMS53 (SCLC), H460, H1155 (LCC) and H1355 (ADC) were previously reported[60]. All the cell lines are mycoplasma-free and were regularly tested for mycoplasma contamination. We regularly authenticate cell lines by testing recognition by autologous CTL clones and HLA-A2 expression when applicable.

The SV40-immortalised human bronchial epithelial cell line 16HBE14o- (16HBE), used as a control (Merck, SCC150).

**QRT-PCR**. Total RNA was immediately extracted from sorted cell populations using the Single Cell RNA Purification Kit (Norgen Biotek) or TRIzol reagent (Invitrogen) for human samples. cDNA was synthesised using the Maxima First Strand cDNA Synthesis Kit (ThermoFischer Scientific). qRT-PCR was performed on a Step-One Plus (Applied Biosystems) using Maxima SYBR Green Master Mix (ThermoFischer Scientific). Expression levels of transcripts were normalised to 18S expression. PCR primers and probes for human (NRP1-2, PLXNA1-4, PLXND1, SEMA-3A-G, 18S) and mouse (Batf, Cd244, Tigit, Egr2, 18S) genes were designed by Sigma-Aldrich and used according to the manufacturer's recommendations. A complete list of all primers is available in Supplementary Table 3.

**Mouse cell lines, tumour engraftment and peptide vaccine**. The B16F10 melanoma cell line (H-2b) and MC-38 colon tumour cell line were purchased from the American Type Culture Collection and Kerafast (ATCC, CRL-6475™ and Kerafast, ENH204-FP, respectively). Tumour cells were grown in DMEM/F-12 medium (ThermoFischer Scientific) supplemented with 10% fetal calf serum (FCS), 2 mM L-glutamine, 1 mM sodium pyruvate, and antibiotics (50 U/ml penicillin and 50 µg/ml streptomycin). Tumour volume was measured using a caliper every 2–3 days and estimated using the following formula: $\pi/6 \times length \times width \times thickness$ (mm³). The maximum tumour size allowed by the institutional ethical board is of 2000 mm³. Criteria for early endpoint are deteriorating body condition score, weight loss, tumour size and ulcerated, necrotic or infected tumours, euthanasia was done by C02 inhalation. We have adhered to these size limits in all experiments.

Female C57BL/6 J mice were purchased from Envigo. For each experiment, groups of 4–10 mice 7–9 weeks of age received $2 \times 10^5$ B16F10 melanoma cells or MC-38 colon cancer cells subcutaneously (s.c.) in the right flank. All animals were housed at Gustave Roussy's animal facility and treated in accordance with guidelines established by the institutional animal committee (CEEA no. 26: 2015–041–1229 and 2018–056–16280).

For the cancer vaccine, C57BL/6 J mice were immunised s.c. with 100 µg of gp100 (KVPRNQDWL) and/or Trp2 (SVYDFFVWL) peptides (GeneCust) plus 25 µg of poly(I:C) adjuvant (InvivoGen), at day 5 and every week at the tail base.

**Murine TIL isolation**. Tumours were harvested at days 8–21 and digested for 40 min at 37 °C according to Tumour Dissociation Kit protocol (Miltenyi Biotec 130–096–730). Tumours were crushed on 100 µm cell strainers and washed twice with PBS 2% FCS. Single-cell suspensions were enriched for CD45+ or CD8+ cells using the MultiMACS system (Miltenyi Biotec). In brief, cells from tumour tissues were labelled with anti-CD45 or anti-CD8a microbeads (Miltenyi Biotec 130–052–301 or 130–117–044, respectively), and then purified using the POSSEL program on MultiMACS. The positive fraction was recovered for TIL analysis by flow cytometry or ex vivo assays.

**Antibodies and flow cytometry**. For human cell surface and intracellular staining, anti-CD3 (clone UCHT1, Biolegend 300424, 1/200), -CD4 (clone RPA-T4, Biolegend 300517, 1/200), -CD8α (clone RPA-T8, Biolegend 301023, 1/200), -CD25 (clone CD25-3G10, Invitrogen MHCD2501, 1/100), -FoxP3 (clone 259D, Biolegend 320213, 1/50), -PD-1 (clone J105, eBioscience 25-2799-42, 1/100) and -Nrp-1

(clone 12C2, Biolegend 354504, 1/25) mAb, and -human IgG-Fc (clone HP6017, Biolegend 409306, 1/100) were used. Cell surface and intracellular staining of mouse cells was performed on single-cell suspensions using antibodies specific to the following molecules: CD3 (clone 17A2, Biolegend 100241, 1/100), CD4 (clone RM4-5, Biolegend 100536, 1/200), CD8α (clone REA601, Miltenyi 130-109-252, 1/20), CTLA-4 (clone UC10-4B9, Miltenyi 130-102-570, 1/20), PD-1 (clone 29 F.1A12, Biolegend 135223, 1/100), Tim-3 (clone RMT3⁻23, Biolegend 119705, 1/100), LAG-3 (clone C9B7W, Biolegend 125211, 1/100), Nrp-1 (clone 3E12, Biolegend 145209, 1/20), T-bet (clone REA102, Miltenyi 130-119-783, 1/50), NFATc1 (clone 7A6, Biolegend 649603, 1/100), Blimp-1 (clone 5E7, Biolegend 150005, 1/100), Helios (clone 22F6, Biolegend 137210, 1/25), IRF-4 (clone REA201, Miltenyi Biotec 130-100-915, 1/20), FoxP3(clone FJK-16s, Thermo Fischer Scientific 25-5773-82, 1/200), Ki-67 (clone REA183, Miltenyi Biotec 130-120-418, 1/50), perforin (clone eBioOMAK-D, eBioscience 17-9392-80, 1/50) and granzyme B (clone GB11, Biolegend 515403, 1/50). Dead cells were excluded using the Live/Dead Fixable Blue Dead Cell Stain Kit (Invitrogen 1866842).

For intracellular staining, cells were fixed/permeabilized with the FoxP3 Staining Buffer Set according to the manufacturer's instructions (eBioscience 00-5523-00). Staining of Trp2 and gp100-specific T cells was performed using H-2-Kb/SVYDFFVWL and H-2-Db/KVPRNQDWL dextramers, respectively (Immudex). Gating strategies to sort or analyse the respective cell population are depicted in Supplementary Figs. 7 and 8. Flow cytometric analysis was conducted on an LSR Fortessa (BD) and analysed using FlowJo software (Tree Star).

**In vitro migration assay**. Freshly isolated CD8+ TIL were incubated for 30 min to 1 h either in medium or in the presence of neutralising anti-Nrp-1 mAb (clone 761704, R&D systems MAB59941, 10 µg/mL), anti-PD-1 (clone RMP1-14, Bio-X-Cell BE0146, 10 µg/mL) or isotype control (clone 2A3, Bio-X-Cell BE0089, 10 µg/ml). B16F10 tumour cells were cultured for 2 days in the lower chambers of Transwell plates (Corning) and then TIL were seeded in the upper chambers to trigger T-cell migration. After 2 h at 37 °C, the number of CD8+ T cells that had migrated into the lower chambers was counted by flow cytometry and phenotyped.

For experiments with freshly isolated human lung TIL and CTL clone P62, activated T cells were incubated for 30 min to 1 h either with BSA (Merck) or Sema-3A-Fc (R&D systems 1250-S3-025, 100 ng/ml), and their ability to migrate toward the human rCXCL12 (Peprotech 300-28 A, 50 nM) was evaluated. Results were expressed as chemotaxis index.

**T-cell stimulation, cytokine production and cytotoxic assay**. Purified CD45+ TIL were co-cultured for 4 h with B16F10 tumour cells, pulsed with Trp2 (1 µg) and gp100 (1 µg) peptides, in the presence of Brefeldin A (Invitrogen 00-4506-51, 1/1000), monensin (Merck M5273, 2 µM) and anti-CD107a mAb (clone 1D4B, Biolegend 121619, 1/50). TIL were stained with mAb specific for surface proteins prior to fixation and permeabilization. Permeabilized cells were then stained with anti-IFNγ (clone XMG1.2, Biolegend 505808, 1/200), anti-TNF (clone MP6-XT22, Biolegend 506321, 1/100) and anti-perforin (clone eBioOMAK-D, eBioscience 17-9392-80, 1/50) mAb.

For cytotoxicity experiments, freshly isolated CD8+ TIL were either kept in medium or pretreated with neutralising anti-Nrp-1 (clone 761704, R&D systems MAB59941, 10 µg/ml), anti-PD-1 (clone RMP1-14, Bio-X-Cell BE0146, 10 µg/ml) or isotype controls (clone 2A3, Bio-X-Cell BE0089, 10 µg/ml). Cytotoxic activity toward the B16F10 cell line, pulsed with Trp2 and gp100 peptides, or the MC-38 cell line was evaluated using a conventional 4 h and overnight chromium ($^{51}$Cr) release assay.

For experiments with human freshly isolated lung TIL and CTL clone P62, activated T cells were incubated for 30 min to 1 h either with bovine serum albumin (BSA, Merck) or Sema-3A-Fc (R&D systems 1250-S3-025, 100 ng/ml), and their cytotoxic activity toward the autologous tumour cell was evaluated.

**Western blot analyses**. Equivalent amounts of protein extracts from tumour cell lines were separated by SDS-PAGE and transferred to a nitrocellulose membrane[61]. Blots were then probed with anti-Sema-3B mAb (clone 904201, R&D systems MAB5440, 1 µg/ml), anti-Sema-3A (clone 215803, R&D systems MAB1250, 1 µg/ml) or anti-β-actin-peroxidase (clone AC-15, Merck A3854, 1/50 000), followed by secondary anti-mouse (Santa cruz Biotechnology sc-2031) or anti-rat (R&D systems HAF005) horseradish peroxidase-conjugated Ab. Anti-F-actin was included as a loading control. Uncropped and unprocessed scans of all western blots are available in the Source Data file.

**In vivo PD-1 and Nrp-1 blockade**. Mice were treated i.p. with 100 µg/mouse of anti-PD-1 (clone RMP1-14, Bio-X-Cell BE0146) mAb and/or i.t. with 25 µg/mouse or i.p. with 75 µg/mouse of anti-Nrp-1 (clone 761704, R&D systems MAB59941) mAb. For tumour outgrowth experiments, mice were treated on days 6, 8, 10, 12 and 14 after tumour inoculation for B16F10 or days 4, 7, 10 and 16 after tumour inoculation for MC-38, and TIL were sorted and analysed on day 17 for B16F10 or day 21 for MC-38.

**Statistical analysis**. GraphPad Prism5 software (GraphPad Software, Inc., San Diego, CA, USA) was used for graphic representation and statistical analysis.

Statistical significance was determined with the one-way or two-way analysis of variance test with Bonferroni correction, or with the two-tailed paired or unpaired Student t test.

**Reporting summary**. Further information on research design is available in the Nature Research Reporting Summary linked to this article.

## Data availability

The authors state that all data generated during this study are included in the article, its supplementary information file, and the Source Data file, and are available from the corresponding author upon reasonable request.

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

## Acknowledgements

We thank Benjamin Besse, Department of medecine Gustave Roussy, for providing lung cancer patient PBMC. We are grateful to all members of Gustave Roussy's animal facility (Plateforme d'Evaluation pré-clinique) for their help with in vivo experiments. We thank the staff of the cytometry facility (Plateforme d'Imagerie-Cytométrie) of Gustave Roussy for flow cytometry analyses. This work was supported by grants from the Association pour la Recherche sur le Cancer (ARC) and the Institut national du Cancer (INCa). ML was a recipient of a MENRT fellowship from the French Ministry of Research, the Ligue contre le Cancer and SIRIC-SOCRATE; SC and EV are supported by a grant from INCa.

## Author contributions

Conception and design: M Leclerc, G Bismuth and F Mami-Chouaib. Development of methodology: M Leclerc and F Mami-Chouaib. Acquisition of data (providing animals, acquiring and managing patients, providing facilities, etc.): M Leclerc, G Bismuth, E Voilin, G Gros, S Corgnac, V de Montpréville, P Validire and F Mami-Chouaib. Analysis and interpretation of data (e.g. statistical analysis, biostatistics, computational analysis): M Leclerc and F Mami-Chouaib. Writing, reviewing and/or revision of the manuscript: M Leclerc, G Bismuth and F Mami-Chouaib. Administrative, technical and material support (i.e. reporting and organising data, constructing databases): M Leclerc, G Gros, E Voilin, V de Montpréville, P Validire and F Mami-Chouaib. Study supervision: F Mami-Chouaib.

## Additional information

**Competing interests:** The authors declare no competing interests.

