## [Peer Review File · Nature Communications]

Reviewers' comments:

Reviewer #1 (T cell exhaustion, anti-tumor T response)(Remarks to the Author):

In the present study Leclerc et al. examined the expression and function of Neuropilin-1 (Nrp-1) in tumor infiltrating T cells. They found Nrp-1 to be expressed in activated TIL and its expression to be correlated with an effector phenotype and with high expression of various activation markers including CD25, CD69, PD-1, CTLA-4, Tim3 and LAG3. Because Nrp-1 is a repulsive molecule, its expression in activated TIL suggested that blockade of its interaction with its natural ligand Sema-3, should allow migration of activated TIL to the tumor site and improved anti-tumor function. This outcome was observed in the experiments that performed by the authors and reported in the present study.

The work has been performed in an organized manner and the results observed in every experiment were consistent with the anticipated outcomes based on the known properties and function of Nrp-1.

Specific points:

- 1) The role of Nrp-1 in CD8+ TIL has been previously studied and reported in mouse and human systems. Several of these previous reports are cited in the present study. Thus, the work of the present manuscript lacks novelty overall and the advancement provided in the field is only incremental.
- 2) Expression of Nrp-1 in human NSCLC was examined by qRT-PCR and expression was normalized to that observed in healthy lung (Supplemental Figure S1). The results of the healthy lung tissue are not shown. Moreover, it seems that only one NSCLC sample (#8) has a compelling increase of Nrp1 mRNA expression compared to healthy lung. Among the other NSCLC samples, two seem to have a 2x expression level of Nrp-1 compared to healthy lung, whereas the remaining five samples seem to have lower expression of Nrp-1 and likely equivalent expression of Nrp-2 to that of healthy lung. These findings diminish the biological significance of the studies and raise few major questions: i) are Nrp-1/2 are indeed expressed in the lung cancer TME? ii) are they expressed in TIL or other cell types? iii) are they expressed in healthy lung and in some cases of NSCLC Nrp-1/2 are elevated in unidentified cell types in the lung? Similar concerns apply regarding SEMA3 expression (Supplemental Figure S1).
- 3) The above concern is further substantiated by the fact that the studies have been performed using one human T cell clone (P62) generated from TIL of one patient with NSCLC. In order to generalize their observations and conclusions, the authors have to investigate whether these results can be reproduced in TIL from several patients with NSCLC not just from one patient.
- 4) Figure 2: Based on the findings that Nrp-1 is expressed in activated P62 cells and prevents their migration toward CXCL12 gradient, the authors concluded that Nrp-1 via its ligand Sema-3A negatively regulates effector function of Nrp-1+ CTL. The results are not truly consistent with an inhibitory effect of Nrp-1 on CTL effector function but only with an effect on the migration of effector CTL. This is a known function of Nrp-1 and this point should be clearly stated.
- 5) Figure 4: The fact that Nrp-1 is co-expressed with PD-1, CTLA4, Tim3 and LAG3, is not indicative of a role of Nrp-1 on the exhaustion mechanism. Instead, this observation is expected as Nrp-1 is expressed only on highly activated T cells similarly to the expression of these inhibitory receptors. In fact, induced expression of these inhibitory receptors and Nrp-1 is part of the physiologic mechanism for downregulation of the immune response of highly activated T cells by upregulation of inhibitory signals and repulsion of these activated T cells from the site of the ongoing immune response. These issues should be corrected and the relevant biological significance of these observations should be carefully stated.
- 6) Figure 5: The authors observed that during ex vivo stimulation Nrp-1+PD-1+ TIL had higher expression of IFN γ -producing cells than their Nrp-1- counterparts and interpreted this observation as an indication that Nrp-1+ cells had a more advanced exhaustion state. This conclusion statement is inconsistent with these findings and is, actually, the reverse from what these results show because

exhausted cells have the lowest capacity for IFN γ production on rechallenge. The present results are indicative of a more activated state of Nrp-1+ cells.

7) The in vitro results of Figure 5d showed that a Nrp-1 or PD-1 blockade increased T cell-mediated lysis but their combination did not have an additive effect. In contrast, the in vivo studies shown in Figure 6 showed that combined injection of Nrp-1 and PD-1 blocking antibodies had an additive effect. What is the mechanism of the differential outcome in these two experimental systems? The authors have not addressed this important point experimentally or even verbally in their discussion by considering tentative mechanistic explanations.

8) In the in vivo experiments the Nrp-1 blocking antibody was injected intratumorally, not systemically as the PD-1 blocking antibody. It should be discussed what was the reason for administration in the tumor site, whether systemic injection of Nrp-1 blocking antibody was tested and what the outcome was in such approach. This is particularly important if the investigators would like to propose that combining Nrp-1 blockade with PD-1 blockade will improve the outcome of anti-PD-1 therapy clinically, especially in tumors such as NSCLC, in which intratumoral injection is not technically feasible.

Minor points:

1) In the result section entitled "Interaction of human Nrp-1 with Sema-3A impairs T-cell effector functions in vitro" in line 6 from the bottom, the authors meant to indicate Fig. 2d instead of Fig. 2c.

2) In the result section entitled "Nrp-1 typifies a highly activated tumour-specific CD8+ TIL subset with impaired functional activities" in line 9 from the bottom, referring to the percentages of MAA-specific T cells in the Nrp-1-PD-1- TIL, the numbers should be 3% and 0% according to the data shown in Figure 5a.

Reviewer #2 (Immune checkpoint blockade, tumor biology)(Remarks to the Author):

The referee compliments with the authors for the quality of their research.

The paper introduce a further player (NRP)-1 at the level of the immune checkpoints for controlling tumour-specific CD8 T-cell functions.

The study might prospect new possible combinations in cancer with the the combination of anti PD1 and (NRP)-1.

There are only minimal comments in order to complete and discuss their research in a more complete manner.

The authors should report if they have data about the combination anti PD1 and (NRP)-1 versus the combination anti PD1 +anti CTLA4

In any case they should briefly comment on the eventual superiority or inferiority of both combinations.

Few words on other systems to block (NRP)-1 effects should be reported:

ie: A homology-based SEMA3C protein structure was created, and its interaction with the neuropilin (NRP)-1 receptor was modeled to guide the development of the corresponding disrupting compounds J Endocr Soc. 2018 Oct 11;2(12):1381-1394. Targeting Semaphorin 3C in Prostate Cancer With Small Molecules.

The reviewer would see at the end of the paper a better organised section on conclusions.

Reviewer #3 (Immune checkpoint blockade, clinical trial)(Remarks to the Author):

This manuscript by Dr. Mami-Chouaib and colleagues gives an in-depth analysis of the role of NRP1 as an immune checkpoint. The manuscript has many strengths. A few comments.

1) In figure 1d, although a p-value <0.05 was not found, there is a suggestion of correlation with Foxp3. It would be good to see some numerical value that would allow assessment of the correlation beyond just $p > 0.05$.

2) The reason for selection of Sema-3A as the sole ligand to study in one assay is not entirely clear from the presented data. Were other ligands evaluated and shown not to have such impact?

3) It is difficult to know the value of minor decreases in the rate of growth of in the animal studies. This should be addressed.

4) The discussion tends to reiterate the findings of the manuscript, rather than addressing limitations or unanticipated events. For instance, one major conclusion is to evaluate with PD-1 inhibitors, but cytotoxicity for instance was interestingly not increased with the combination. Description of why the decrement in tumor growth is more relevant would be helpful.

Reviewers' comments:

Reviewer #1 (T cell exhaustion, anti-tumor T response)

(Remarks to the Author):

In the present study Leclerc et al. examined the expression and function of Neuropilin-1 (Nrp-1) in tumor infiltrating T cells. They found Nrp-1 to be expressed in activated TIL and its expression to be correlated with an effector phenotype and with high expression of various activation markers including CD25, CD69, PD-1, CTLA-4, Tim3 and LAG3. Because Nrp-1 is a repulsive molecule, its expression in activated TIL suggested that blockade of its interaction with its natural ligand Sema-3, should allow migration of activated TIL to the tumor site and improved anti-tumor function. This outcome was observed in the experiments that performed by the authors and reported in the present study.

The work has been performed in an organized manner and the results observed in every experiment were consistent with the anticipated outcomes based on the known properties and function of Nrp-1.

Specific points:

1) The role of Nrp-1 in CD8⁺ TIL has been previously studied and reported in mouse and human systems. Several of these previous reports are cited in the present study. Thus, the work of the present manuscript lacks novelty overall and the advancement provided in the field is only incremental.

Although Nrp-1 has been previously reported to be expressed by mouse and human melanoma-infiltrating CD8⁺ T lymphocytes, its role in regulating CD8⁺ T-cell functions and its potential inhibitory effects on anti-tumour T-cell activities has never been studied. This is now stated in the introduction of our manuscript (page 4). Our work is innovative because we demonstrate for the first time that Nrp-1 expression on CD8 T lymphocytes interferes with their cytotoxic activity (Fig. 5 d), production of the lymphocyte pore-forming protein perforin (Fig. 5f) and migratory potential (Fig. 5g), and that its blockade is able to restore these T-cell effector functions. Similar conclusions were obtained with the colon tumour model MC-38 (Fig. 6i and 6j), and human TIL freshly isolated from NSCLC tumours (Fig. 2e and 2g) and the CTL clone P62 (Fig. 2d and 2f).

2) Expression of Nrp-1 in human NSCLC was examined by qRT-PCR and expression was normalized to that observed in healthy lung (Supplemental Figure S1). The results of the healthy lung tissue are not shown. Moreover, it seems that only one NSCLC sample (#8) has a compelling increase of Nrp1 mRNA expression compared to healthy lung. Among the other NSCLC samples, two seem to have a 2x expression level of Nrp-1 compared to healthy lung, whereas the remaining five samples seem to have lower expression of Nrp-1 and likely equivalent expression of Nrp-2 to that of healthy lung. These findings diminish the biological significance of the studies and raise few major questions: i) are Nrp-1/2 are indeed expressed in the lung cancer TME? ii) are they expressed in TIL or other cell types? iii) are they expressed in healthy lung and in some cases of NSCLC Nrp-1/2 are elevated in unidentified cell types in the lung? Similar concerns apply regarding SEMA3 expression (Supplemental Figure S1).

Relative expression of Nrp-1 in human NSCLC was first examined by qRT-PCR, and expression was normalized to that in autologous healthy lung tissues (Supplemental Figure S1). Values obtained with healthy lungs were not shown because they were used as references and therefore they were all equal to 1. As requested by the reviewer, we now included results of all healthy lung tissues in a new version of Supplementary Fig. 1.

Nrp-1 has been reported to be expressed by endothelial cells, dendritic cells, Treg cells as well as several other normal cells and malignant cells (1-8). Using specific mAb, we checked Nrp-1 expression at the protein level on tumour cells and on several NSCLC tumour cell lines. Results show that Nrp-1 is expressed on freshly isolated human lung tumour cells from three NSCLC patients (Supplementary Fig. 2a) and on several NSCLC tumour cell lines (Supplementary Fig. 2b). Thus, and to answer the reviewer's question, the molecule can be expressed by TIL but also by many other cells types, including cancer cells. This is now more clearly stated in the manuscript (page 6) and several references are now included to mention this point (17-23). We agree that 4 out of 8 NSCLC samples analyzed by qRT-PCR for Nrp-1 expression in Supplementary Fig. 1a were negative compare to their autologous healthy tissue, but we do not see why this could diminish the significance of the study since our work is focused on the role of Nrp-1 in T-cell responses and since, in the numerous samples of NSCLC studied (see Fig. 1a), the molecule was found up-regulated in TIL. So, and from a "therapeutic point of view", we are convinced that it could be a highly relevant target.

Similarly, Sema-3 expression was analyzed at the mRNA level by RT-PCR. Results indicated that expression of Sema 3 family members varies from one tumour tissue to another (Supplementary Fig. 1c). This is now stated in the results section (page 6). Because anti-human Sema3A and anti-human Sema-3F mAb are available, we tested the expression levels of both proteins by FACS (page 7). Results indicated that Sema-3A (Supplementary Fig. 3a and 3b) and Sema-3F (Supplementary Fig. 3c and 3d) are expressed in some lung tumour tissues (Supplementary Fig. 3a and 3c) and several tumour cell lines (Supplementary Fig. 3b and 3d). Because tumour cells produce more frequently Sema-3A and because rSema-3A-Fc molecule is available, this semaphorin 3 family member was used in further studies.

3) The above concern is further substantiated by the fact that the studies have been performed using one human T cell clone (P62) generated from TIL of one patient with NSCLC. In order to generalize their observations and conclusions, the authors have to investigate whether these results can be reproduced in TIL from several patients with NSCLC not just from one patient.

To further support the conclusion that interaction of Nrp-1 with its ligand interferes with T-cell functions, we examined the consequences of Sema-3A-Fc ligation to Nrp-1 on the migratory behavior of freshly isolated CD8⁺ TIL from three NSCLC patients. Results showed that the interaction of human Nrp-1 with its soluble ligand Sema-3A-Fc inhibits T-cell migration toward CXCL12 (Fig. 2e). Moreover, Sema-3A-Fc ligation to Nrp-1 on freshly isolated polyclonal NSCLC TIL inhibited cytotoxic activity toward autologous fresh tumour cells (Fig. 2g).

4) Figure 2: Based on the findings that Nrp-1 is expressed in activated P62 cells and prevents their migration toward CXCL12 gradient, the authors concluded that Nrp-1 via its ligand Sema-3A negatively regulates effector function of Nrp-1+ CTL. The results are not truly consistent with an inhibitory effect of Nrp-1 on CTL effector function but only with an effect

on the migration of effector CTL. This is a known function of Nrp-1 and this point should be clearly stated.

In Figure 2d, we showed that the interaction of Nrp-1 with its ligand Sema-3A negatively regulates the migratory potential of CTL. Moreover, results included in Figure 2e (Fig. 2f in the new version of figures) show that ligation of Sema-3A-Fc to Nrp-1 negatively regulates the cytotoxic activity of the CTL clone as well as the cytotoxic activity of freshly isolated NSCLC TIL (Fig. 2g). These points are now clearly stated in the manuscript (page 8).

5) Figure 4: The fact that Nrp-1 is co-expressed with PD-1, CTLA4, Tim3 and LAG3, is not indicative of a role of Nrp-1 on the exhaustion mechanism. Instead, this observation is expected as Nrp-1 is expressed only on highly activated T cells similarly to the expression of these inhibitory receptors. In fact, induced expression of these inhibitory receptors and Nrp-1 is part of the physiologic mechanism for downregulation of the immune response of highly activated T cells by upregulation of inhibitory signals and repulsion of these activated T cells from the site of the ongoing immune response. These issues should be corrected and the relevant biological significance of these observations should be carefully stated.

To further demonstrate the involvement of Nrp-1 in the exhaustion state of T cells, we measured intracellular expression of perforin in murine TIL stimulated with B16F10 tumour cells in the absence and the presence of anti-Nrp-1 mAb (page 12). Results shown in Figure 5f indicated that anti-Nrp-1 mAb induced increase in perforin production by Nrp-1⁺PD-1^{hi} CD8⁺ TIL stimulated *ex vivo* with autologous tumour cells.

As mentioned by the reviewer, we stated that upregulation of Nrp-1 inhibitory signal promotes repulsion of activated T cells from the site of the ongoing immune response and downregulation of their functional activities, which is part of the physiologic mechanisms used by the immune system to shutdown specific T-cell immunity (page 10). This is a novel finding never reported before for CD8⁺ TIL and antitumor T-cell response.

6) Figure 5: The authors observed that during *ex vivo* stimulation Nrp-1⁺PD-1⁺ TIL had higher expression of IFN γ -producing cells than their Nrp-1⁻ counterparts and interpreted this observation as an indication that Nrp-1⁺ cells had a more advanced exhaustion state. This conclusion statement is inconsistent with these findings and is, actually, the reverse from what these results show because exhausted cells have the lowest capacity for IFN γ production on rechallenge. The present results are indicative of a more activated state of Nrp-1⁺ cells.

During *ex vivo* stimulation, Nrp-1⁺PD-1⁺ TIL included higher percentages of IFN γ -producing cells than their Nrp-1⁻ counterparts. We agree with the reviewer that this does not indicate that Nrp-1⁺ T cells had a more advanced exhaustion state, but that these cells displayed a more activated state. This statement is now correctly included in the results section (page 11).

7) The *in vitro* results of Figure 5d showed that a Nrp-1 or PD-1 blockade increased T cell-mediated lysis but their combination did not have an additive effect. In contrast, the *in vivo* studies shown in Figure 6 showed that combined injection of Nrp-1 and PD-1 blocking antibodies had an additive effect. What is the mechanism of the differential outcome in these two experimental systems? The authors have not addressed this important point

experimentally or even verbally in their discussion by considering tentative mechanistic explanations.

We agree with the reviewer that combining anti-Nrp-1 plus anti-PD-1 does not have an additive effect on T cell-mediated lysis *in vitro* as compared with Nrp-1 or PD-1 blockade in contrast to our *in vivo* studies. We believe that the mechanism of this differential outcome is likely associated with the capacity of anti-Nrp-1 to enhance T-cell migratory capacity *in vivo* and thus recruitment of TIL at the tumor site without improving CTL activity. Consistently, an increase in the number CD8⁺ T cells/mg of tumour was observed *in vivo* with anti-Nrp-1 plus anti-PD-1 combination but not with each mAb used alone. Moreover, results included in Figure 6i show that TIL from mice treated with anti-Nrp-1 plus anti-PD-1 mediated stronger cytotoxic activity toward MC-38 tumour cells than TIL from mice treated with each mAb alone (pages 14 & 15). These points are now discussed in the discussion section (pages 19 and 20).

8) In the *in vivo* experiments the Nrp-1 blocking antibody was injected intratumorally, not systemically as the PD-1 blocking antibody. It should be discussed what was the reason for administration in the tumor site, whether systemic injection of Nrp-1 blocking antibody was tested and what the outcome was in such approach. This is particularly important if the investigators would like to propose that combining Nrp-1 blockade with PD-1 blockade will improve the outcome of anti-PD-1 therapy clinically, especially in tumors such as NSCLC, in which intratumoral injection is not technically feasible.

In our *in vivo* experiments, anti-Nrp-1 blocking antibody was injected intratumorally (i.t.) and the anti-PD-1 blocking antibody was injected intraperitoneally (i.p.). The reason for the administration of anti-Nrp-1 at the tumour site was to reduce the quantity of Ab needed to neutralize Nrp-1 on TIL and thus to reduce potential side effects because Nrp-1 is also expressed by other normal cells. This statement is now discussed in the manuscript (page 13). We did not test the systemic injection of anti-Nrp-1 blocking antibodies. However, as suggested by the referee and to propose to combine Nrp-1 blockade with PD-1 blockade to improve the outcome of anti-PD-1 therapy in NSCLC patients, we performed *in vivo* experiments where both anti-Nrp-1 and anti-PD-1 blocking antibodies were injected intraperitoneally (page 14). Results indicated that i.p. administration of both mAb induced a better inhibition of tumour growth than administration of mAb in different sites (anti-PD-1: i.p. and anti-Nrp-1: i.t.) (Fig. 6c and d)

Minor points:

1) In the result section entitled “Interaction of human Nrp-1 with Sema-3A impairs T-cell effector functions *in vitro*” in line 6 from the bottom, the authors meant to indicate Fig. 2d instead of Fig. 2c.

In this section, page 8, we replaced Fig. 2c by Fig. 2d.

2) In the result section entitled “Nrp-1 typifies a highly activated tumour-specific CD8⁺ TIL subset with impaired functional activities” in line 9 from the bottom, referring to the

percentages of MAA-specific T cells in the Nrp-1-PD-1- TIL, the numbers should be 3% and 0% according to the data shown in Figure 5a.

In this section, page 11, referring to the percentages of MAA-specific T cells in the Nrp-1-PD-1- TIL, the numbers included take into account all the results shown in the right panel of Figure 5a ($0.9\% \pm 0.3$ and $0.5\% \pm 0.3$), not only the experiment shown in the left panel of this figure indicating 3% and 0%. Accordingly, the percentages of MAA-specific CD8⁺ TIL in the Nrp-1⁺PD-1^{hi} T-cell subset ($8.8\% \pm 1.6$ and $10.4\% \pm 2.6$) and the Nrp-1⁻PD-1⁺ ($2.9\% \pm 0.7$ and $5.6\% \pm 1.2$) TIL subset are from the right panel of Fig. 5a (not the prototype experiment shown in the left panel).

Reviewer #2 (Immune checkpoint blockade, tumor biology)
(Remarks to the Author):

The referee compliments with the authors for the quality of their research.

The paper introduce a further player (NRP)-1 at the level of the immune checkpoints for controlling tumour-specific CD8 T-cell functions.

The study might prospect new possible combinations in cancer with the the combination of anti PD1 and (NRP)-1.

There are only **minimal comments** in order to complete and discuss their research in a more complete manner.

1) The authors should report if they have data about the combination anti PD1 and (NRP)-1 versus the combination anti PD1+anti CTLA4

In any case they should briefly comment on the eventual superiority or inferiority of both combinations.

1) We did not previously perform experiments combining anti-PD-1 with anti-CTLA-4. However, as suggested by the reviewer, we now compared the combination of anti-PD-1 plus anti-Nrp-1 to that of anti-PD-1 plus anti-CTLA-4. Results indicated that both antibody combinations induced delay in tumour growth (Supplementary Fig. 6e) These data are now mentioned in the manuscript (page 14).

2) Few words on other systems to block (NRP)-1 effects should be reported:

ie: A homology-based SEMA3C protein structure was created, and its interaction with the neuropilin (NRP)-1 receptor was modeled to guide the development of the corresponding disrupting compounds
J Endocr Soc. 2018 Oct 11;2(12):1381-1394. Targeting Semaphorin 3C in Prostate Cancer With Small Molecules.

2) As requested by the reviewer, we added in the discussion section a paragraph describing studies performed by Lee CCW et al showing that inhibition of Sema-3C binding to Nrp-1 with small molecules attenuates prostate cancer growth (J Endocr Soc. 2018 Oct 11;2(12):1381-1394) (page 19).

3) The reviewer would see at the end of the paper a better organised section on conclusions.

3) As required by the referee, we added a concluding paragraph at the end of the manuscript to emphasize the therapeutic potential of anti-Nrp-1 (pages 20 & 21).

Reviewer #3 (Immune checkpoint blockade, clinical trial)
(Remarks to the Author):

This manuscript by Dr. Mami-Chouaib and colleagues gives an in-depth analysis of the role of NRP1 as an immune checkpoint. The manuscript has many strengths.

A few comments:

1) In figure 1d, although a p-value <0.05 was not found, there is a suggestion of correlation with Foxp3. It would be good to see some numerical value that would allow assessment of the correlation beyond just $p > 0.05$.

1) In figure 1d, we added the p-value ($p=0.2578$) that suggests the absence of correlation between Nrp-1 expression and Foxp3 expression in a subset of CD4⁺ TIL. As suggested by the reviewer we also added a supplementary Table with numerical values that allow assessment of the lack of correlation (Supplementary Table I). This is now mentioned in the results section (page 7).

2) The reason for selection of Sema-3A as the sole ligand to study in one assay is not entirely clear from the presented data. Were other ligands evaluated and shown not to have such impact?

2) To define the Nrp-1 ligand that could be used in functional assays, we performed immunofluorescence and western blot analyses of several lung tumor cell lines with two available mAbs specific of human Sema-3A and Sema-3F. Results showed that human NSCLC cell lines, including IGR-Pub, express Sema-3A and Sema-3F at the protein level (Supplementary Fig. 3b and 3d). However, since the soluble Nrp-1 ligand available is Sema-3A-Fc and most of tumour cells express it at high level, we used this ligand in the presented experiments. This is now explained in the manuscript (pages 7 and 8).

3) It is difficult to know the value of minor decreases in the rate of growth of in the animal studies. This should be addressed.

We agree with the reviewer that it is difficult to know the values of minor decreases in the rate of tumor growth in *in vivo* studies. This is now addressed (page 13) and a table was included to summarize the obtained results (Supplementary Table II).

4) The discussion tends to reiterate the findings of the manuscript, rather than addressing limitations or unanticipated events. For instance, one major conclusion is to evaluate with PD-1 inhibitors, but cytotoxicity for instance was interestingly not increased with the combination. Description of why the decrement in tumor growth is more relevant would be helpful.

As requested by the reviewer, we now discussed more deeply our results in the discussion section in particular our unexpected data. In this regard, we discussed the effect of anti-Nrp-1 plus anti-PD-1 on specific cytotoxic activity *ex vivo* and on tumor growth and the plausible explanation of these limitations (pages 19 and 20). As requested by referee 2, we also added a

concluding paragraph at the end of the manuscript to discuss the therapeutic potential of anti-Nrp-1 in combination with anti-PD-1 (pages 19 and 20).

REVIEWERS' COMMENTS:

Reviewer #1 (Remarks to the Author):

The authors have addressed my previous concerns, performed additional work and revised their manuscript accordingly. I do not have additional comments on the revised version of the manuscript.

Reviewer #2 (Remarks to the Author):

I consider the revision acceptable from my side.

Reviewer #3 (Remarks to the Author):

I would like to thank the authors for their thoughtful response to the reviews. I feel like the additional experiments have added to the strength of the manuscript. My concerns from the original review have been sufficiently addressed in the revision.